# Cyclin-dependent Kinase 1 and Aurora Kinase choreograph mitotic storage and redistribution of a growth factor receptor

**Christina D. Cota**[1], **Matthew S. Dreier**[2], **William Colgan**[3], **Anna Cha**[4], **Twan Sia**[5], **Brad Davidson**[5]\*

**1** Department of Biology, Colby College, Waterville, Maine, United States of America, **2** NYU Grossman School of Medicine, New York, New York, United States of America, **3** Broad Institute, Massachusetts Institute of Technology, Cambridge, Massachusetts, United States of America, **4** Department of Molecular & Cellular Biology, Harvard University, Cambridge, Massachusetts, United States of America, **5** Department of Biology, Swarthmore College, Swarthmore, Pennsylvania, United States of America

\* bdavids1@swarthmore.edu

**Data Availability Statement:** All relevant representative or numerical data is provided in figures or supplementary data files. Raw image files are stored on Google Drive and access to

## Abstract

Endosomal trafficking of receptors and associated proteins plays a critical role in signal processing. Until recently, it was thought that trafficking was shut down during cell division. Thus, remarkably, the regulation of trafficking during division remains poorly characterized. Here we delineate the role of mitotic kinases in receptor trafficking during asymmetric division. Targeted perturbations reveal that Cyclin-dependent Kinase 1 (CDK1) and Aurora Kinase promote storage of Fibroblast Growth Factor Receptors (FGFRs) by suppressing endosomal degradation and recycling pathways. As cells progress through metaphase, loss of CDK1 activity permits differential degradation and targeted recycling of stored receptors, leading to asymmetric induction. Mitotic receptor storage, as delineated in this study, may facilitate rapid reestablishment of signaling competence in nascent daughter cells. However, mutations that limit or enhance the release of stored signaling components could alter daughter cell fate or behavior thereby promoting oncogenesis.

## Introduction

Dividing cells undergo dynamic shifts in membrane trafficking. During mitotic entry, internalization of plasma membrane promotes cell rounding [1]. As cells exit mitosis, targeted recycling promotes formation of the cytokinetic furrow [2,3]. Membrane and associated integral membrane proteins are trafficked through a well-delineated system of endosomal compartments [1,4]. In this endosomal trafficking network, Rab GTPases dictate compartment-specific functions (Fig 1A). Newly endocytosed vesicles fuse to form early endosomes distinguished by RAB4 and RAB5. These early endosomes can either recycle back to the plasma membrane through RAB4-dependent fast recycling or mature into late endosomes through a RAB7-dependent pathway. Recycling can also occur through a slow, RAB11-dependent pathway. Late endosomes eventually fuse with lysosomes leading to degradation of integral membrane proteins and other cargo [5]. Recent studies have provided some insights into trafficking during mitotic exit, including a key role for RAB11-dependent effectors during assembly of the

these files will be provided on request, by
contacting C.D.C. at cdcota@colby.edu.

**Funding:** C.D.C. was supported by her American
Heart Association Postdoctoral Award
(16POST27250075). All authors were supported
by the National Science Foundation (NSF) (grant
#1656571 awarded to B.D.) Some of the student's
summer salaries along with purchase of some of
their research supplies were funded by
Swarthmore College. The funders had no role in
study design, data collection and analysis, decision
to publish, or preparation of the manuscript.

**Competing interests:** The authors have declared
that no competing interests exist.

**Abbreviations:** AurK, Aurora Kinase; CDK1, Cyclin-
dependent Kinase 1; CDKI, Cyclin-dependent
Kinase Inhibitor; Ceslr1, Cadherin EGF LAG seven-
pass G-type receptor 1; ESCRT, endosomal sorting
complexes required for transport; FGFR, Fibroblast
Growth Factor Receptor; GFP, green fluorescent
protein; HOPS, homotypic fusion and protein
sorting; ORF, open reading frame; PFA,
paraformaldehyde; PLK, Polo-like Kinase; SEM,
standard error of mean; TGF-β, transforming
growth factor beta; TVC, Trunk ventral cell/Cranial-
cardiac progenitor.

cytokinetic furrow [3,6–8]. However, trafficking during mitotic entry remains poorly characterized. Bulk internalization during entry appears to be mediated by suppression of recycling rather than an increase in endocytosis, but the specific endocytic pathways involved in entry trafficking have not been identified [1].

Recent studies have begun to reveal essential roles for mitotic membrane trafficking in tissue homeostasis and embryonic patterning. In the mammalian epidermis, symmetrically dividing cells internalize Ceslr1 (Cadherin EGF LAG seven-pass G-type receptor 1) and other membrane proteins involved in planar cell polarization. Unbiased redistribution of these internalized proteins during mitotic exit appears to be critical for reintegration of dividing cells into the epithelium [9]. In the developing wing of *Drosophila* embryos, unbiased redistribution of transforming growth factor beta (TGF-β) receptors internalized during mitotic entry ensures proper patterning [10]. In a range of asymmetrically dividing embryonic and stem cell lineages, integral membrane proteins involved in Notch signaling are internalized during mitosis [11–13]. Biased redistribution of these signaling components during mitotic exit underlies asymmetric fate specification. Despite the importance of mitotic trafficking in embryonic patterning and tissue integrity, insights into the regulatory hierarchy choreographing the uptake and redistribution of signaling components remain extremely limited.

Mitosis is choreographed by 3 major classes of mitotic kinases, CDK1, Aurora Kinases (AurKs), and Polo-like Kinases (PLKs) [14]. During mitotic entry, these kinases regulate a diverse set of cellular processes required for spindle assembly, centrosome dynamics, chromatid separation, and cytokinesis. As cells progress through metaphase, Cyclin B is degraded, and the resulting loss of CDK1 activity is critical for promoting exit-specific cellular processes. Remarkably, due in part to the long-standing assumption that trafficking was shut down during mitosis [2,3,15–17], very few studies have addressed the regulatory roles of these kinases in mitotic trafficking. Exceptions include research on the mammalian epidermis demonstrating that PLK1-dependent phosphorylation of the planar cell polarity protein Celsr1 mediates mitotic internalization [18]. In yeast, PLK1 has also been reported to phosphorylate ESCRT (endosomal sorting complexes required for transport) proteins required for septation [19]. Poor characterization of the regulatory links between mitotic kinases and division-specific trafficking patterns represents a fundamental gap in our understanding of the interplay between cell division and signaling.

We have begun to address this gap by studying cranial-cardiac progenitor specification in the invertebrate chordate, *Ciona intestinalis* (Type A, also referred to as *Ciona robusta*). In *Ciona* embryos, the heart is derived from a set of 4 precardiac founder cells. Each founder cell divides asymmetrically to produce 1 cranial-cardiac progenitor (or trunk ventral cell, TVC) and 1 tail muscle progenitor (Fig 1A; [20,21]). Fibroblast Growth Factor (FGF) receptors are unequally distributed during founder cell division [20]. Immediately following division, differential inheritance of FGF receptors generates asymmetric FGF-dependent induction of cranial-cardiac progenitor cell fate [21–23]. Localized cell–matrix adhesion biases mitotic FGFR redistribution through localized retention and/or recycling of Caveolin-rich membrane domains and associated FGF receptors [20]. By characterizing the regulation of mitotic FGFR redistribution in *Ciona* founder cells, we aim to reveal more general mechanisms for mitotic trafficking and explore how these mechanisms are biased during asymmetric divisions.

## Results

### FGF receptor distribution patterns during founder cell mitosis

We precisely quantified mitotic FGFR redistribution through volumetric analysis (Fig 1). These assays were conducted using *Mesp>FGFR::Venus* transgenic embryos [21]. Because the Mesp enhancer specifically drives transgene expression in the heart founder cell lineage, we

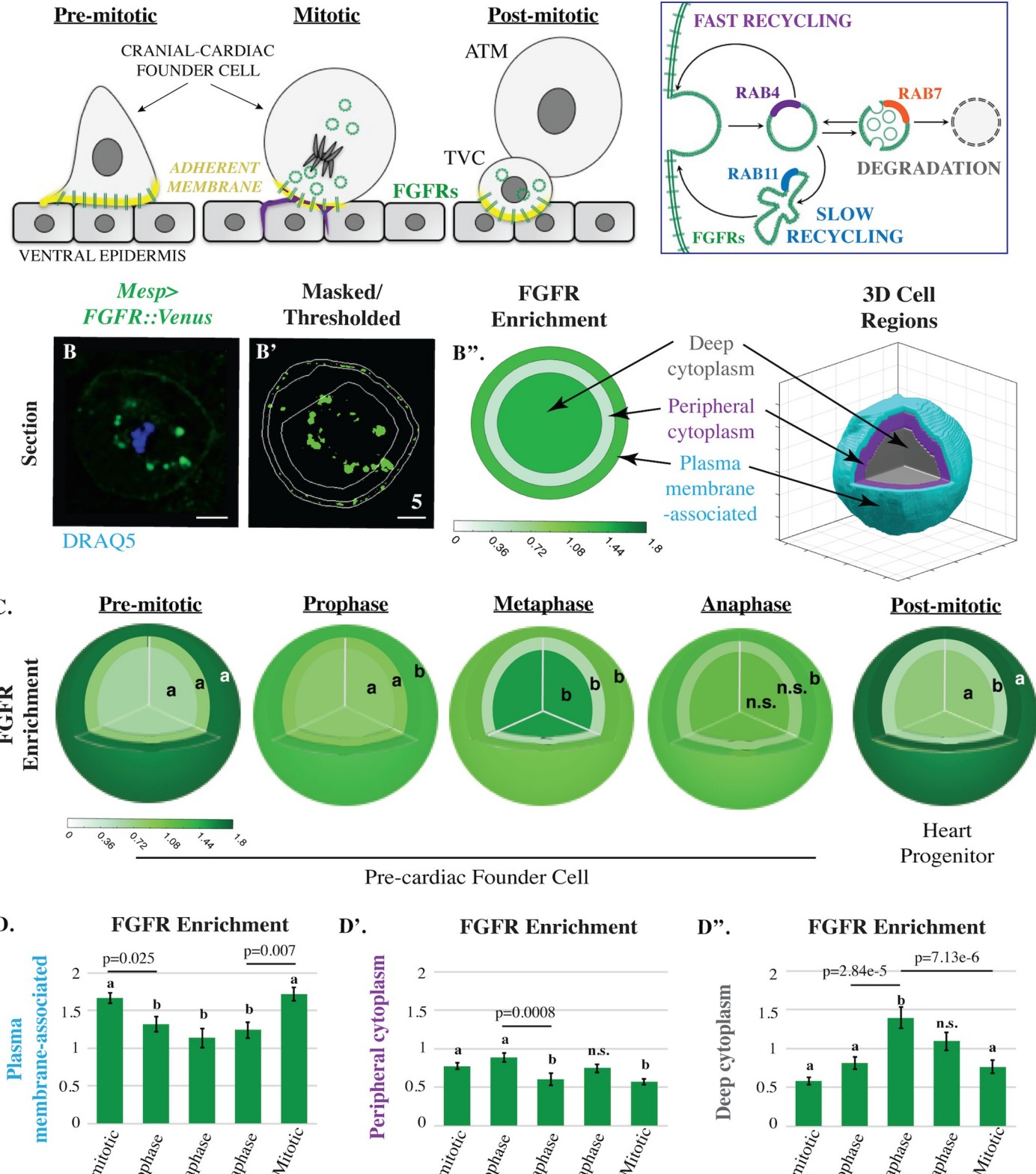

**A. Differential FGFR Redistribution During Founder Cell Division**

**Fig 1. Mitotic trafficking of FGF receptors during founder cell division.** (A) Models depicting differential FGFR (green) redistribution during asymmetric founder cell division based on previous data (left panel) [20,41] along with a summary of endosomal pathways (right panel). For simplicity, schematics depict lateral views of a single founder cell. Regions of actin enrichment (purple; [40].) and adherent membrane (yellow, [41]) are indicated. (B-B") Transverse sections and graphical summary depicting 3D-volumetric analysis of FGFR::VENUS distribution (quantified as regional enrichment; Methods) in a representative mitotic founder cell. Lines indicate region boundaries (white). Scale bars are indicated in micrometers. (C-D) Diagrammatic and graphical summaries of regional FGFR::VENUS enrichment (green) during founder cell division. Some regions are labeled with an a or b to denote that significant changes ($p < 0.05$) occurred within this region across cell cycle stages. Other regions are labeled n.s. to denote that no significant changes occurred for the indicated stages. Sample numbers for each stage are as follows: premitotic $n = 50$, prophase $n = 36$, metaphase $n = 17$, anaphase $n = 24$, and post-mitotic $n = 34$. Significance was determined using one-way ANOVA followed by Tukey multiple comparison test. Numerical values for all graphs can be found in S1 Data. ATM, Anterior Tail Muscle Cell; FGFR, Fibroblast Growth Factor Receptor; TVC, Trunk ventral cell/Cranial-cardiac progenitor.

are able to analyze FGFR::VENUS distribution in vivo [20]. Thus, transverse sections (such as Fig 1B) represent confocal stacks of mitotic founder cells that were dividing within intact embryos (as illustrated in Fig 1A). Distribution patterns of transgenically expressed FGFR::VENUS were assessed in 3 concentric regions (plasma membrane, peripheral cytoplasm, and deep cytoplasm, Fig 1B–1B"; Methods). FGFR::VENUS expression in founder cells is very low, precluding live imaging analysis (see Fig 3I"). Instead, transgenic *Mesp>FGFR::Venus* embryos were fixed at 15-min intervals spanning founder cell mitosis and costained with an anti-green fluorescent protein (GFP) antibody to visualize FGFR::VENUS and a chromatin marker (DRAQ5) to facilitate precise mitotic staging [20,22]. Volumetric analysis provided a rigorous and highly reproducible measurement of FGFR::VENUS distribution at each cell cycle stage that is not well represented by individual transverse sections. Thus, in this and subsequent figures, we focus on providing a complete set of graphical data (Fig 1C and 1D) rather than representative confocal sections for each stage (Fig 1B). Through this analysis, we identified 3 significant, stage-specific shifts in FGFR distribution (Fig 1C and 1D). As founder cells entered prophase, FGFR enrichment along the plasma membrane was dramatically reduced. As cells progressed into metaphase, FGFR enrichment shifted from the peripheral to the deep cytoplasm. Thus, during mitotic entry, FGFR-enriched membranes were gradually internalized. During mitotic exit, this trend was reversed as FGFR enrichment shifted from the deep cytoplasm to the plasma membrane-associated region. Our quantitative analysis demonstrates that FGFR distribution tightly correlates with mitotic progression. Critically, these mitotic patterns of FGFR distribution are highly reproducible within stage-matched cells providing a robust framework for further experimental analysis.

In order to determine whether FGFR redistribution is a mitotically regulated process, we blocked founder cell division through targeted overexpression of a *Ciona* ortholog to Cyclin-dependent Kinase Inhibitor/p27 (*Mesp>Cdki-b/p27*; [23,24]). Founder cells expressing CDKI-b and FGFR::VENUS were fixed approximately 1 hour after control cells complete asymmetric division (Hotta Stage 16; [22]). CDKI-b expression induced interphase arrest and blocked FGFR internalization (S1 Fig). Indeed, the FGFR distribution pattern in CDKI-b–expressing founder cells at Stage 16 closely matched that of premitotic controls (Hotta Stage 14; S1 Fig). Notably, arrested founder cells tended to undergo cranial-cardiac cell fate induction, indicating that mitotic internalization is not required for inductive signaling (S1 Fig). These results indicate that temporal correlations between FGFR distribution patterns and mitotic stage reflect a functional, regulatory relationship.

## Endosomal pathways involved in mitotic redistribution of FGFR

We next began to investigate the endosomal pathways associated with each stage-specific shift in FGFR distribution (Fig 2). Each shift correlated with discrete changes in colocalization between labeled FGFR (FGFR::VENUS) and markers of late endosomes (CLIP::RAB7, Fig 2A–2E") or slow recycling endosomes (CLIP::RAB11, Fig 2D and 2F–2F"). In contrast, no

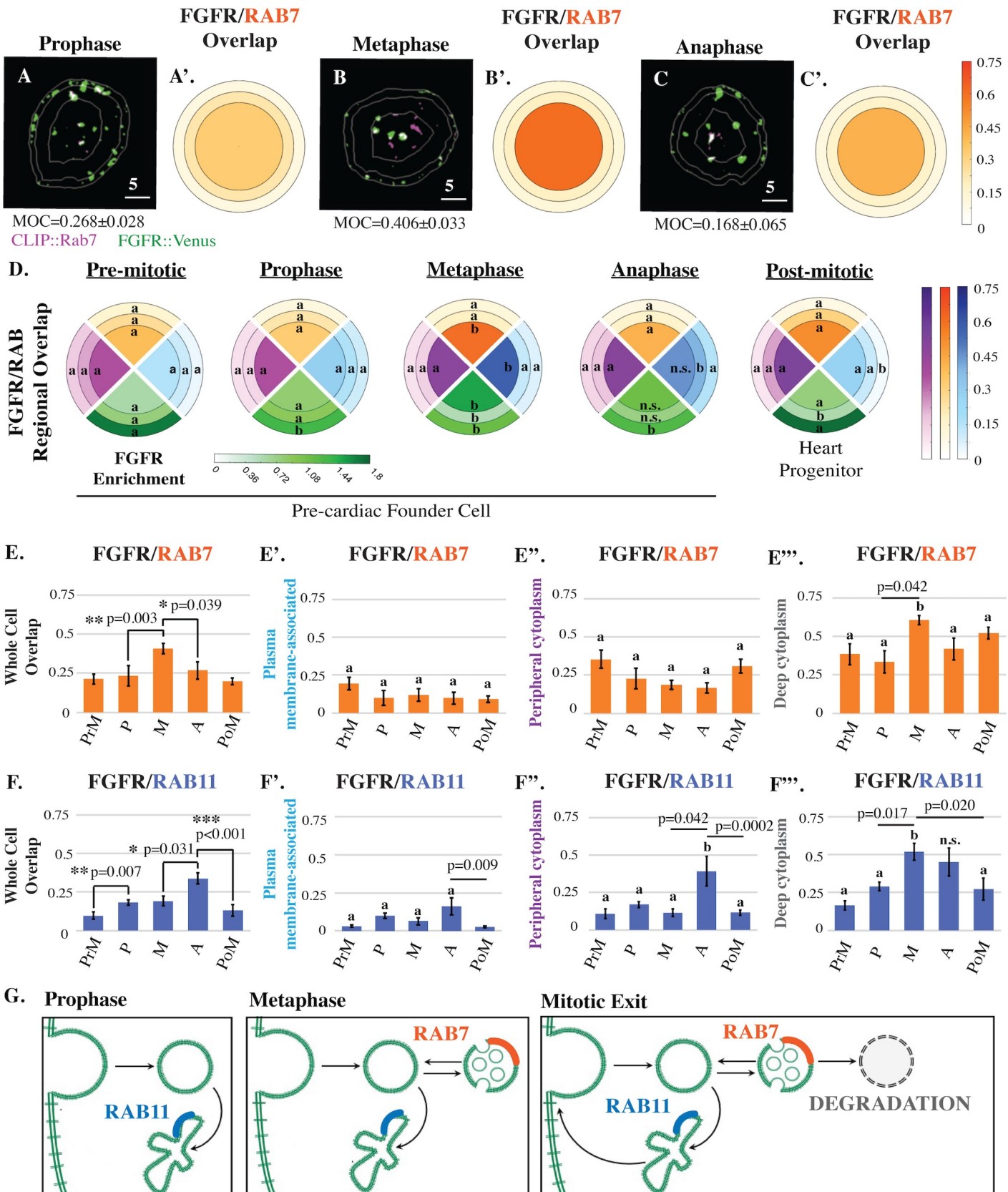

**Fig 2. Mitotic FGFR trafficking during founder cell division. (A-C')** Masked/thresholded transverse sections and graphical summary depicting 3D-volumetric analysis of FGFR::VENUS/CLIP::RAB-GTPase colocalization (Manders' overlap; MOC) in representative mitotic founder cells. Lines indicate region boundaries (white). Scale bars are indicated in micrometers. **(D)** Graphical summary of regional FGFR::VENUS enrichment (green) and FGFR::VENUS/ CLIP::RAB-GTPase colocalization (CLIP::RAB4, purple; CLIP::RAB7, orange; CLIP::RAB11, blue) during founder cell division. Some regions are labeled with an a or b to denote that significant changes ($p < 0.05$) occurred within this region across stages. Other regions are labeled n.s. to denote that no significant changes occurred for the indicated stages. **(E-F')** Quantification of total (whole cell; **E** and **F**) and regional (**E'-E'''**, **F'-F'''**) FGFR::VENUS/ CLIP::RAB-GTPase colocalization during founder cell division showing significant changes in RAB7 and RAB11 values. Significance was determined using one-way ANOVA followed by Tukey multiple comparison test. Numerical values for all graphs can be found in S2 Data. **(G)** Model of mitotic FGFR trafficking illustrating stage-specific shifts as indicated. **See also S2 Fig.** FGFR, Fibroblast Growth Factor Receptor; MOC, Manders' overlap coefficient.

significant changes were observed in colocalization with a marker of fast recycling endosomes (CLIP::RAB4; Fig 2D, S2 Fig). During prophase, whole cell colocalization between labeled FGFR and RAB11 increased (Fig 2D and 2F). As cells entered metaphase, whole cell and deep cytoplasmic FGFR/RAB7 colocalization increased (Fig 2A, 2B', 2D and 2E). During this phase, FGFR/RAB11 whole cell colocalization remained stable (Fig 2F), but there was a significant increase in deep cytoplasmic enrichment (Fig 2D and 2F'''). This regional shift in RAB11 colocalization may reflect trafficking of existing FGFR-containing recycling endosomes toward the spindle poles [23]. During anaphase, whole cell and deep cytoplasmic FGFR/RAB7 colocalization decreased, while whole cell and peripheral FGFR/RAB11 colocalization increased (Fig 2A–2F'''). As cells exited division, whole cell and peripheral FGFR/RAB11 colocalization decreased (Fig 2D and 2F–2F''). Taken together, our colocalization data support a 3-part model for mitotic FGFR trafficking (Fig 2G). FGF receptors are first internalized and stored in slow recycling endosomes during prophase. During metaphase, stored receptors are either retained in slow recycling endosomes or shunted to a maturation pathway. During mitotic exit, receptors stored in slow recycling endosomes are returned to the plasma membrane, while receptors stored in late endosomes are either recycled or degraded.

## CDK1 suppresses FGFR degradation during mitotic entry

We next sought to investigate the role of the primary mitotic entry kinase, CDK1, in FGFR trafficking (Fig 3). By treating late gastrulae (Hotta Stage 14) with a fast-acting CDK1 inhibitor (roscovitine/seliciclib), we were able to block CDK1 activity in mitotic founder cells. Through DRAQ5 staining, we were able to identify treated founder cells displaying chromatin condensation, indicating that they had been arrested in prophase. We began investigating the impact of this treatment on FGFR trafficking using transgenic *Mesp>FGFR::Venus*, *Mesp>CLIP::Rab7* embryos. Intriguingly, arrested founder cells displayed a dramatic decrease in FGFR::VENUS staining (Fig 3A, 3B' and 3E). To investigate whether the observed reduction in FGFR::VENUS was a nonspecific result of mitotic arrest, we treated founder cells with AurK inhibitors (Aurora A/B inhibitor: VX-680 or Pan-Aurora Kinase inhibitor: AMG-900). Because these drugs act relatively slowly, we treated embryos just prior to founder cell division (Hotta Stage 13). Treatment with either inhibitor at this stage resulted in prophase arrest, but there was no discernable reduction in FGFR::VENUS staining (VX680, Fig 3C–3C'). We used the same assay to examine the impact of roscovitine treatment on another integral membrane protein, E-CADHERIN::GFP (*Mesp>E-Cadherin::GFP*; Fig 3F–3H). In contrast with the FGFR::VENUS results, roscovitine treatment had no discernable impact on E-CADHERIN::GFP staining. Thus, it appears that CDK1 stabilizes a subset of membrane proteins during mitotic entry rather than having a global, nonspecific impact.

Based on these results, we hypothesized that CDK1 activity promotes FGFR storage by suppressing lysosomal degradation. To test this hypothesis, we inhibited lysosomal degradation in founder cells through targeted expression of a dominant-negative form of the homotypic fusion and protein sorting (HOPS) complex subunit VAM2 (*Mesp>HALO::Vam2*[421-841];

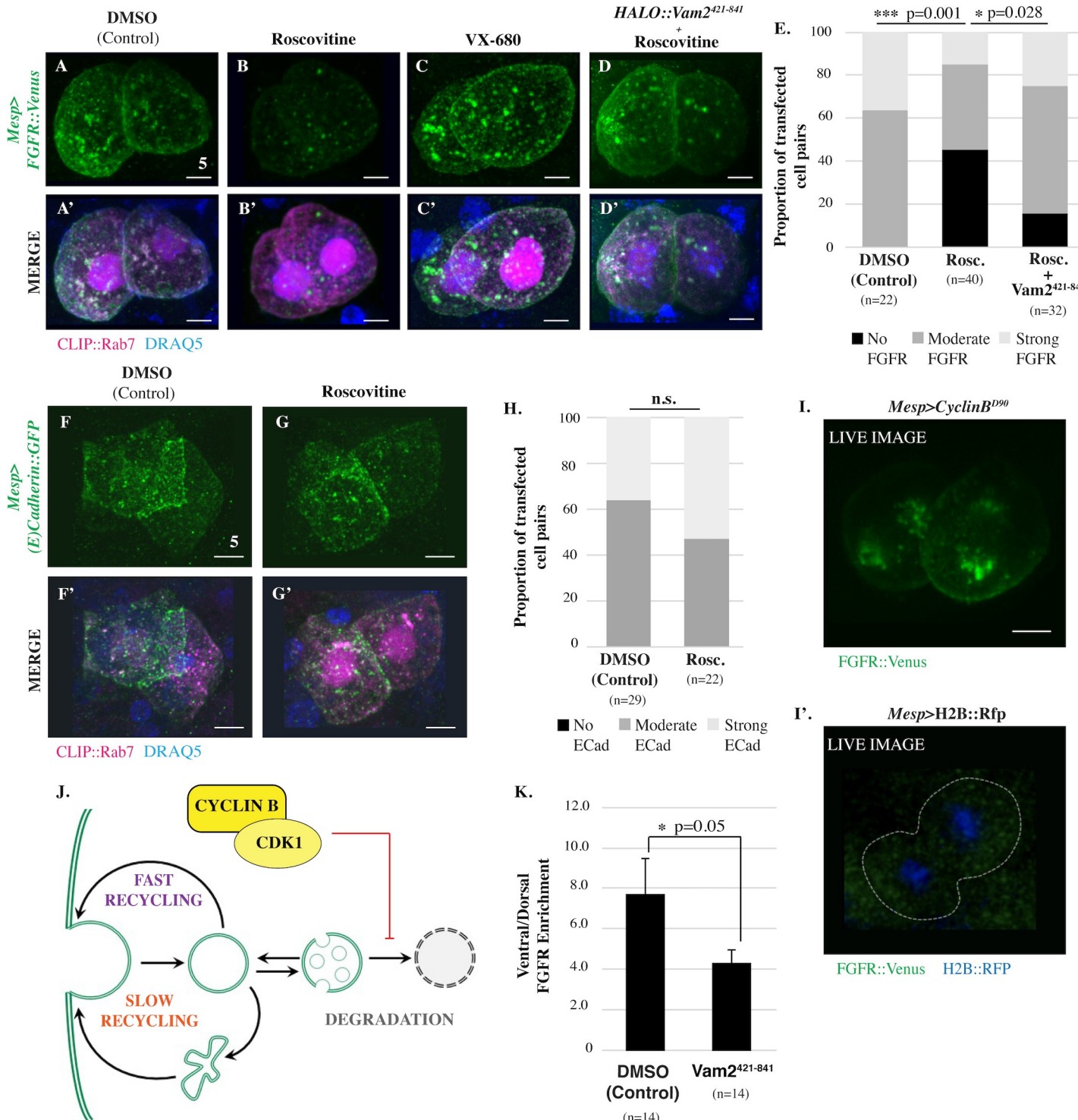

**Fig 3. CDK1 inhibits lysosomal degradation of FGFR.** (**A-D'**) Ventral projections of founder cell pairs electroporated with *Mesp>FGFR::Venus* alone or in combination with *Mesp>HALO::Vam2*[421-841] as indicated and treated with vehicle (DMSO), Roscovitine (14 μmol/L) or VX-680 (21 μmol/L). In this experiment, *Mesp>CLIP::Rab7* was included as a positive control for transfection. (**E**) Qualitative scoring of FGFR::VENUS intensity in transfected founder cell pairs. Significance was determined using Fisher exact test followed by Pearson chi-squared test. n = number of founder cell pairs scored. Treatment with AMG-900 (10 μmol/L) also had no significant impact on FGFR::VENUS intensity. Indeed, there was a nonsignificant increase in the number of cell pairs displaying strong FGFR::VENUS signal in the treated samples—45.8% ± 4.17 of AMG-900-treated cell pairs (*n* = 21) versus 34.5% ± 1.04 of DMSO-treated cell pairs (*n* = 42), *p* = 0.523. (**F-G'**) Ventral projections of founder cell pairs electroporated with *Mesp>E-Cadherin::GFP* and treated with vehicle (DMSO), or roscovitine (14 μmol/L). *Mesp>CLIP::Rab7* was included as a

positive control for transfection. (**H**) Qualitative scoring of E-CADHERIN::GFP intensity in transfected founder cell pairs. No significant differences found between treatments indicated. Significance was determined using Fisher exact test followed by Pearson chi-squared test. n = number of founder cell pairs scored. (**I-I'**) Ventral projection of FGFR::VENUS distribution in transgenic representative live founder cell pairs coelectroporated with *Mesp>FGFR::Venus* and *Mesp>CyclinB^{Δ90}* (**I**) or a control coelectroporated with *Mesp>FGFR::Venus* and *Mesp>H2B::RFP* (**I'**). Note that GFP/YFP signal in the heart founder lineage in the control (outlined by a white dashed line) are not above background levels. This image is representative of numerous observations of *Mesp>FGFR::Venus* in live embryos in which it is impossible to discern any signal leading to the standard use of antibody staining in fixed samples to assay FGFR localization. (**J**) Model depicting proposed CDK1-dependent inhibition of FGFR::VENUS degradation. (**K**) Quantification of FGFR::VENUS polarization in founder cells electroporated and treated as indicated. n = number of founder cells analyzed. Significance was determined using one-way ANOVA followed by Tukey multiple comparison test. Numerical values for all graphs can be found in S3 Data. Scale bars are indicated in micrometers. **See also S3 Fig.** CDK1, Cyclin-dependent Kinase 1; FGFR, Fibroblast Growth Factor Receptor; GFP, green fluorescent protein; YFP, yellow fluorescent protein.

[24]). As predicted by our hypothesis, Vam2^{421-841} expression restored FGFR::VENUS staining in roscovitine-treated samples (Fig 3D–3D' and 3E). We also tested this hypothesis through a gain of function assay involving targeted expression of truncated Cyclin B (*Mesp>Cyclin B^{Δ90}*). Because CYCLIN B^{Δ90} cannot be targeted for degradation by the anaphase-promoting complex, expression of this protein leads to sustained CDK1 activity and inhibits mitotic exit [25]. Despite high levels of transgene expression, observation of FGFR::VENUS in wild-type founder cells requires antibody staining, presumably due to low abundance of the fusion protein [20]. Expression of Cyclin B^{Δ90} led to a dramatic increase in FGFR::VENUS signal, allowing direct observation of FGFR::VENUS in live, unstained embryos (Fig 3I). As seen previously, no FGFR::VENUS signal was detected in live, matched controls (Fig 3I'). Taken together, these results indicate that CDK1 activity suppresses lysosomal FGFR degradation during mitotic entry (Fig 3J). We also treated embryos with roscovitine during interphase. As predicted by our model, this treatment had no discernable impact on FGFR::VENUS staining. Interestingly, Vam2^{421-841} expression disrupted ventral FGFR::VENUS enrichment (Fig 3K). This result suggests that lysosomal degradation contributes to the biased redistribution of internalized FGFRs during asymmetric founder cell division.

## CDK1-dependent phosphorylation of RAB4 suppresses FGFR recycling

We next investigated whether CDK1 regulates other aspects of FGFR trafficking. The recovery of FGFR::VENUS staining in transgenic *HALO:Vam2^{421-841}* embryos allowed us to perform endocytic pathway colocalization analysis in roscovitine-treated cells. While roscovitine treatment had no discernable impact on regional FGFR::VENUS/CLIP::RAB11 colocalization (S3 Fig), we did observe a significant decrease in FGFR::VENUS/CLIP::RAB4 colocalization in peripheral cytoplasm and plasma membrane-associated regions (Fig 4A–4C, S3 Fig). This result suggests that CDK1 activity disrupts the delivery of FGFR-enriched fast recycling endosomes to the plasma membrane along with the shedding of Rab4 which occurs during this process (Fig 4D, [26]). As predicted by this hypothesis, sustained CDK1 activity resulting from transgenic expression of *Cyclin B^{Δ90}* dramatically reduced FGFR::VENUS enrichment along the plasma membrane and promoted robust enrichment of this protein at the spindle poles (S4 Fig). This hypothesis was also supported by a robust increase in the plasma membrane-associated enrichment of FGFR::VENUS in *HALO::Vam2^{421-841}* cells treated with roscovitine in comparison to DMSO controls (S5 Fig). These results indicate that CDK1 suppresses the RAB4-dependent fast recycling pathway and thereby promotes accumulation of internalized FGF receptors (Fig 4D). This model aligns with previous studies indicating that bulk internalization of the plasma membrane during mitotic entry involves decreased recycling rates while internalization rates remain constant [1,27].

We next began to examine the molecular mechanism by which CDK1 impacts the fast recycling pathway. In mammalian cells, CDK1-dependent phosphorylation of RAB4 leads to dissociation of RAB4 from endosomal membranes [28]. However, the impact of this phosphorylation event on mitotic receptor trafficking has not been previously examined. We

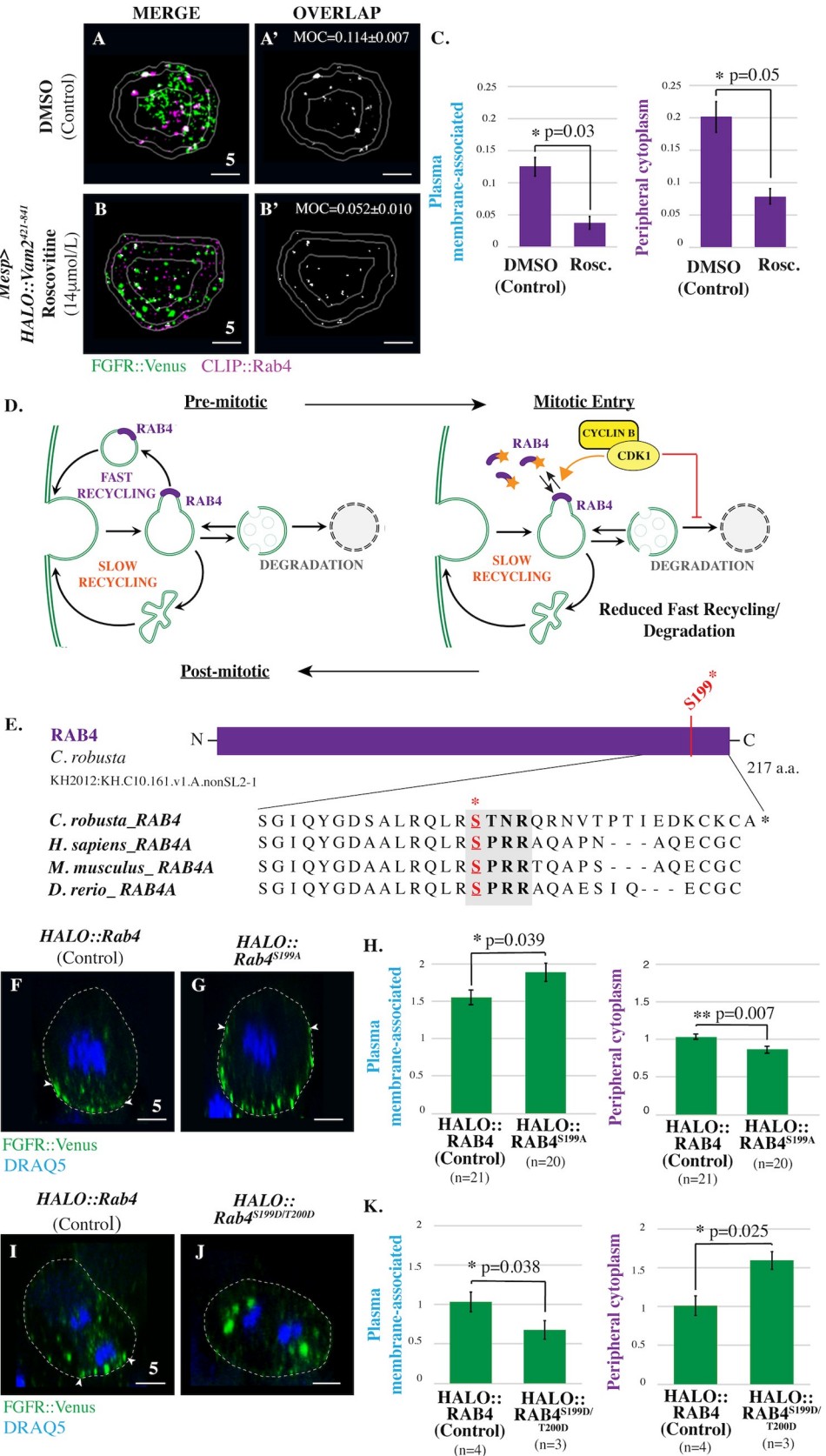

FGFR::Venus    CLIP::Rab4

FGFR::Venus
DRAQ5

**Fig 4. CDK1 inhibits RAB4-dependent fast recycling of FGFR during mitotic entry.** (**A-B'**) Masked/thresholded transverse sections of founder cells electroporated with *Mesp>FGFR::Venus* alone or in combination with *Mesp>HALO::Vam2$^{421-841}$* and treated with vehicle (DMSO) or Roscovitine (14 μmol/L) as indicated. For clarity, images showing only the colocalized FGFR::VENUS/ CLIP::RAB-GTPase puncta in representative sections are provided (OVERLAP; Manders' overlap; MOC) (**A'** and **B'**). (**C**) Quantification of regional FGFR::VENUS/CLIP:: RAB4 colocalization for founder cells electroporated and treated as indicated. (**D**) Model depicting proposed CDK1-dependent regulation of FGFR::VENUS trafficking. (**E**) Schematic depiction of *C. robusta* RAB4 protein. ClustalW alignment shows conservation of previously reported CDK1 phosphorylation motif (bold; [27]). Red asterisk indicates the serine residue phosphorylated by CDK1 in human cells. Putative phosphorylated serine residues in orthologs are indicated (S, red). (**F-G**) Lateral sections of prophase founder cells electroporated with either *Mesp>FGFR::Venus* along with either *Mesp>HALO::Rab4* or *Mesp>HALO::Rab4$^{S199A}$* as indicated. (**H**) Quantification of regional FGFR::VENUS enrichment in prophase founder cells electroporated as indicated. n = number of founder cells analyzed. (**I-J**) Lateral sections of anaphase founder cells electroporated with either *Mesp>FGFR::Venus* along with either *Mesp>HALO::Rab4* or Mesp>HALO::Rab4$^{S199D/T200D}$ as indicated. (**K**) Quantification of regional FGFR::VENUS enrichment in anaphase founder cells electroporated as indicated. n = number of founder cells analyzed. Significance was determined using one-way ANOVA followed by Tukey multiple comparison test (**C, H, K**). Numerical values for all graphs can be found in S4 Data. Dashed lines indicate cell membranes that were delineated by phalloidin staining (**F-G** and **I-J**; red). White arrowheads (**F-G** and **I-J**) indicate dorsal boundaries of membrane-associated FGFR::VENUS puncta. Scale bars are indicated in micrometers. **See also S4 Fig, S5 Fig and S6 Fig.** CDK1, Cyclin-dependent Kinase 1; FGFR, Fibroblast Growth Factor Receptor; MOC, Manders' overlap coefficient.

hypothesized that CDK1-dependent RAB4 phosphorylation suppresses recycling (Fig 4D). The previously reported CDK1 phosphorylation site in RAB4 is highly conserved across vertebrate and invertebrate chordate taxa (Fig 4E). Thus, we were able to test this hypothesis through targeted expression of phospho-deficient forms of *Ciona* RAB4 in which the putative CDK1 phosphorylation site has been mutated (*Mesp>HALO::Rab4$^{S199A}$*). As predicted by our hypothesis, founder cell-specific expression of phospho-deficient Rab4 (*Mesp>HALO:: Rab4$^{S199A}$*) led to increased enrichment of FGFR::VENUS along the plasma membrane during prophase (Fig 4F–4H). We also observed a complementary reduction in FGFR::VENUS enrichment in the peripheral cytoplasm. To determine whether CDK1-dependent phosphorylation was sufficient to inhibit RAB4-dependent recycling of FGF receptors, we generated a phospho-mimetic RAB4 (*Mesp>HALO::Rab4$^{S199D/T200D}$*). As predicted by our hypothesis, *Mesp>HALO::RAB4$^{S199D/T200D}$* appeared to block recycling during mitotic exit, leading to the accumulation of large FGFR-containing puncta in the deep cytoplasm during anaphase (Fig 4I–4K). Quantitative analysis revealed a significant increase in FGFR::VENUS enrichment in the peripheral cytoplasm complemented by significantly reduced enrichment at the plasma membrane. To explore the impact of CDK1-dependent regulation of RAB4 on FGF-dependent induction of the cranio-cardiac progenitor lineage, we coelectroporated embryos with *Mesp>Ensc::GFP* to label all founder lineage cells, *FoxF>RFP* to label cranio-cardiac progenitors along with either phospho-deficient *Mesp>HALO::RAB4$^{S199A/T200A}$*, phospho-mimetic *Mesp>HALO::RAB4$^{S199D/T200D}$*, or a control construct (*Mesp>LacZ* or *Mesp>HALO::RAB4*). As predicted by our model, expression of phospho-deficient RAB4 resulted in a significant increase in cranial-cardiac progenitor induction, while expression of phospho-mimetic RAB4 resulted in a significant decrease in cranial-cardiac progenitor induction (S6 Fig). Taken together, these results indicate that the previously characterized CDK1-dependent phosphorylation of RAB4 serves to inhibit receptor recycling during mitotic entry (Fig 4D). Additionally, these results indicate that CDK1-mediated inhibition of receptor recycling can modulate subsequent cell fate decisions.

## Aurora Kinase suppresses slow recycling of FGFR containing endosomes

We next examined the role of AurK in mitotic FGFR trafficking. The *Ciona* genome contains a single ortholog for AurK (Aurora A/B; [29]). As mentioned previously, treatment with AurK

inhibitors (VX-680 and AMG-900) did not reduce FGFR::VENUS staining (Fig 3C and 3C'). Instead we observed that inhibitor treatment led to a dramatic and significant increase in FGFR::VENUS enrichment in the plasma membrane-associated region (Fig 5A–5C, S7 Fig). Based on this result, we hypothesized that AurK blocks delivery of FGFR from RAB11 slow recycling endosomes to the plasma membrane during mitotic entry, complementing inhibition of the RAB4-dependent fast recycling pathway by CDK1. In line with this hypothesis, we found that inhibitor treatment also significantly decreased FGFR::VENUS/CLIP::RAB11 whole cell colocalization in prophase arrested cells (Fig 5D–5I', S7 Fig). In contrast, these inhibitors had no discernable impact on FGFR::VENUS/CLIP::RAB4 colocalization (S7 Fig) and had variable and contradictory impacts on FGFR/RAB7 colocalization (S7 Fig). Our results indicate that CDK1 and AurK work in tandem to promote storage of internalized FGF receptors during mitotic entry, suppressing both fast and slow recycling pathways (Fig 5K).

## Discussion

Based on our data, we propose a new model for mitotic regulation of FGFR trafficking (Fig 6). According to our model, CDK1 and AurK synergize to promote FGFR storage during mitotic entry. We propose that mitotic receptor storage generates 2 functionally discrete pools. One pool consists of FGF receptor-enriched vesicles shunted into either fast or slow recycling pathways. CDK1 and AurK maintain this pool by suppressing recycling pathways. The second pool consists of FGFR-enriched vesicles that have been shunted into the maturation pathway. CDK1 maintains this pool by suppressing degradation. As cells exit division, the associated inactivation of CDK1 releases both pools of accumulated receptors. Reinitiation of fast recycling restores receptor enrichment on the plasma membrane. Reinitiation of degradation may bias this process, leading to nonuniform receptor redistribution. In *Ciona* founder cells, it appears that matrix adhesion polarizes FGFR trafficking during mitosis, leading to elevated receptor accumulation on the nascent heart progenitor membrane and differential induction (Fig 1A, [20]). Our current model posits that integrin-dependent enrichment of caveolin within adhesive membranes dictates polarized FGFR trafficking [20]. Current studies are focused on determining the specific contributions of integrin and caveolin to this process. We are also investigating whether adhesion suppresses FGFR degradation or promotes FGFR recycling, thereby biasing the redistribution of "stored" FGFR during mitotic exit (Fig 6). Previous studies suggest that PLK1-mediated activation of slow recycling may also contribute to receptor recycling and/or receptor redistribution [18,19]. We are currently exploring whether this conserved role for PLK1 overcomes AurK-dependent suppression of slow recycling (Fig 2D and 2F') to promote delivery of FGF receptors from the RAB11 recycling compartment to the plasma membrane as illustrated in our model (Fig 6).

Our data also suggest that the previously characterized CDK1-dependent phosphorylation of RAB4 [28] directly suppresses RAB4-mediated recycling during mitotic entry (Fig 4E–4K). This regulatory relationship appears to be broadly conserved as indicted by sequence conservation of the CDK1 phosphorylation site across a wide range of vertebrate Rab4a genes, along with orthologous genes from a variety of tunicates (including *Ciona*, Fig 4E) amphioxus, echinoderms, mollusks, cnidarians, and even potentially in slime molds. Interestingly, this phosphorylation site does not appear to be conserved in vertebrate Rab4b genes, although they do contain a potential alternate CDK1 phosphorylation site. We also hypothesize that CDK1-dependent suppression of fast recycling may be a general feature of mitotic entry deployed in a wide variety of cell types to reduce membrane surface area during mitotic rounding which leads, incidentally, to storage and sequestration of associated membrane proteins. Indeed, this

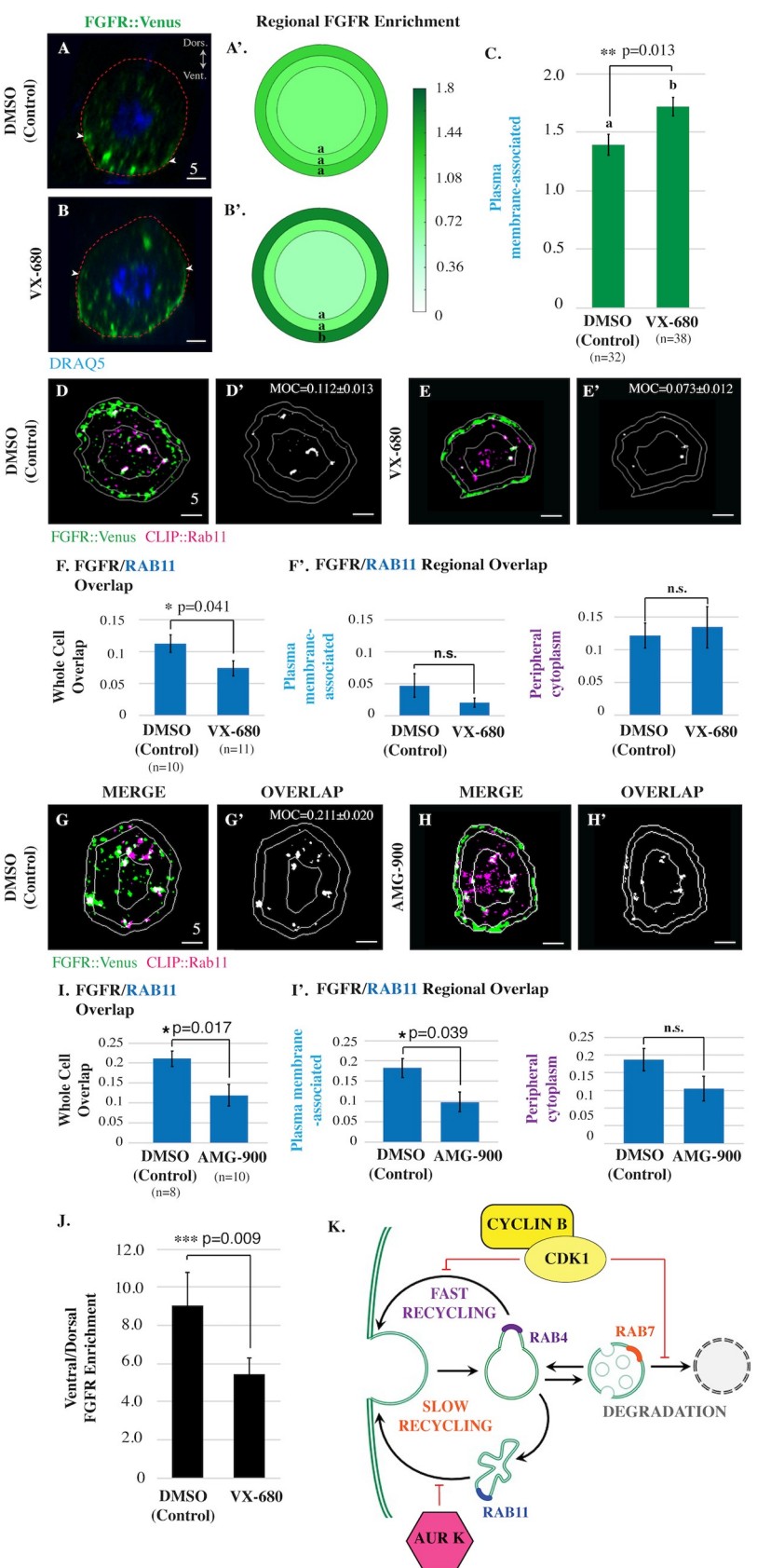

**Fig 5. AurK promotes endosomal maturation and inhibits slow recycling of FGFR during mitotic entry.** (**A-C**) Lateral sections, graphical summary, and quantitative analysis of regional FGFR::VENUS enrichment for founder cells electroporated with *Mesp>FGFR::Venus* and treated with vehicle (DMSO) or VX-680 (21 µmol/L) as indicated. n = number of founder cells analyzed. (**D-E**) Masked/thresholded transverse sections of founder cells electroporated with *Mesp>FGFR::Venus* and *Mesp>HALO::RAB11*. For clarity, images showing only colocalized FGFR::VENUS/ CLIP::RAB-GTPase puncta in representative sections are provided (OVERLAP; Manders' overlap; MOC) (**D'** and **E'**). (**F**) Graphical summary of total (whole cell) or regional FGFR::VENUS/CLIP::RAB11 colocalization (Manders' overlap). (**G-H**) Masked/thresholded transverse sections of founder cells electroporated with *Mesp>FGFR::Venus* and *Mesp>HALO::RAB7*. For clarity, images showing only colocalized FGFR::VENUS/ CLIP::RAB-GTPase puncta in representative sections are provided (MOC for panel H' = 0.119±0.027) (**G'** and **H'**). (**I**) Graphical summary of total (whole cell) or regional FGFR::VENUS/CLIP::RAB7 colocalization (Manders' overlap). (**J**) Quantification of FGFR::VENUS ventral/dorsal polarization in founder cells treated with vehicle (DMSO) or VX-680 (21 µmol/L) as indicated. Significance was determined using one-way ANOVA followed by Tukey multiple comparison test (**C, F, I, J**). Numerical values for all graphs can be found in S5 Data. (**K**) Proposed model of CDK1 and AurK-dependent regulation of mitotic FGFR::VENUS trafficking during mitotic entry. In all micrographs, red dashed lines indicate cell membranes as delineated by phalloidin staining. Scale bars are indicated in micrometers. White arrowheads (**A-B**) indicate dorsal boundaries of membrane-associated FGFR::VENUS puncta. **See also S7 Fig.** AurK, Aurora Kinase; CDK1, Cyclin-dependent Kinase 1; FGFR, Fibroblast Growth Factor Receptor; MOC, Manders' overlap coefficient.

hypothesis aligns with previous data showing that reduced recycling rates during mitotic entry are required for mitotic cell rounding [1]. Intriguingly, the CDK1 phosphorylation motif in Rab4 (SPKK; Fig 4E) is a hotspot for cancer-associated mutations (https://www.cbioportal.org; [30,31]). Future work will investigate whether these specific mutations impact receptor trafficking in dividing cells. We are also currently exploring whether other key regulatory nodes in our model involve direct interactions between the mitotic kinases and Rab GTPases or if they involve a more complex regulatory circuit. For instance, AurK may suppress slow recycling through direct phosphorylation of—Rab11, Rab 11 effectors such as FIPs, Myosin VB, the kinesin Kif13A, or the exocyst subunit EXOC6 [4,32]. Alternatively, AurK may directly disrupt downstream factors associated with delivery of slow recycling endosomes to the plasma membrane including Arf6 and its effectors [33]. It is also possible that AurK suppresses slow recycling through an indirect mechanism similar to documented cascades involved in AurK-dependent regulation of cytokinesis [34,35]. Additionally, our data indicate that CDK1 specifically suppresses degradation for a subset of membrane proteins, including FGFR, rather than uniformly influencing the degradation pathway (Fig 3F–3H). We are currently investigating the range of receptors subjected to the mitotic trafficking pathways we have identified and the molecular basis for this selectivity.

Mitotic receptor storage, as delineated in this study, poses a number of potential benefits and risks. Suppression of lysosomal degradation may facilitate retention of signaling components allowing daughter cells to rapidly reacquire signaling competence. In asymmetrically dividing cells, stored receptors can be rapidly redistributed in response to polarized intrinsic or extrinsic cues generating robust asymmetry in nascent daughter cells. Furthermore, mitotic internalization may serve to sequester receptors during the dynamic process of cell division and prevent spurious signaling. Cell rounding during mitotic entry entails extensive remodeling of the cell membrane and actin cortex along with disassembly of cell–cell and cell–matrix adhesions [36]. Thus, signal modulation provided by membrane microdomains [37,38] or by extensive cross-talk between adhesion and signaling complexes [39] are compromised in dividing cells. Moreover, alterations in cell composition and morphology associated with tissue growth and repair can dramatically alter the signaling environment of dividing cells exacerbating the potential for signal misinterpretation. Thus, sequestration of growth factor receptors during division may play a key role in the suppression of unintended signaling and associated oncogenic behaviors. Conversely, mutations that lead to precocious release of stored receptors could reverse this sequestration and promote oncogenesis.

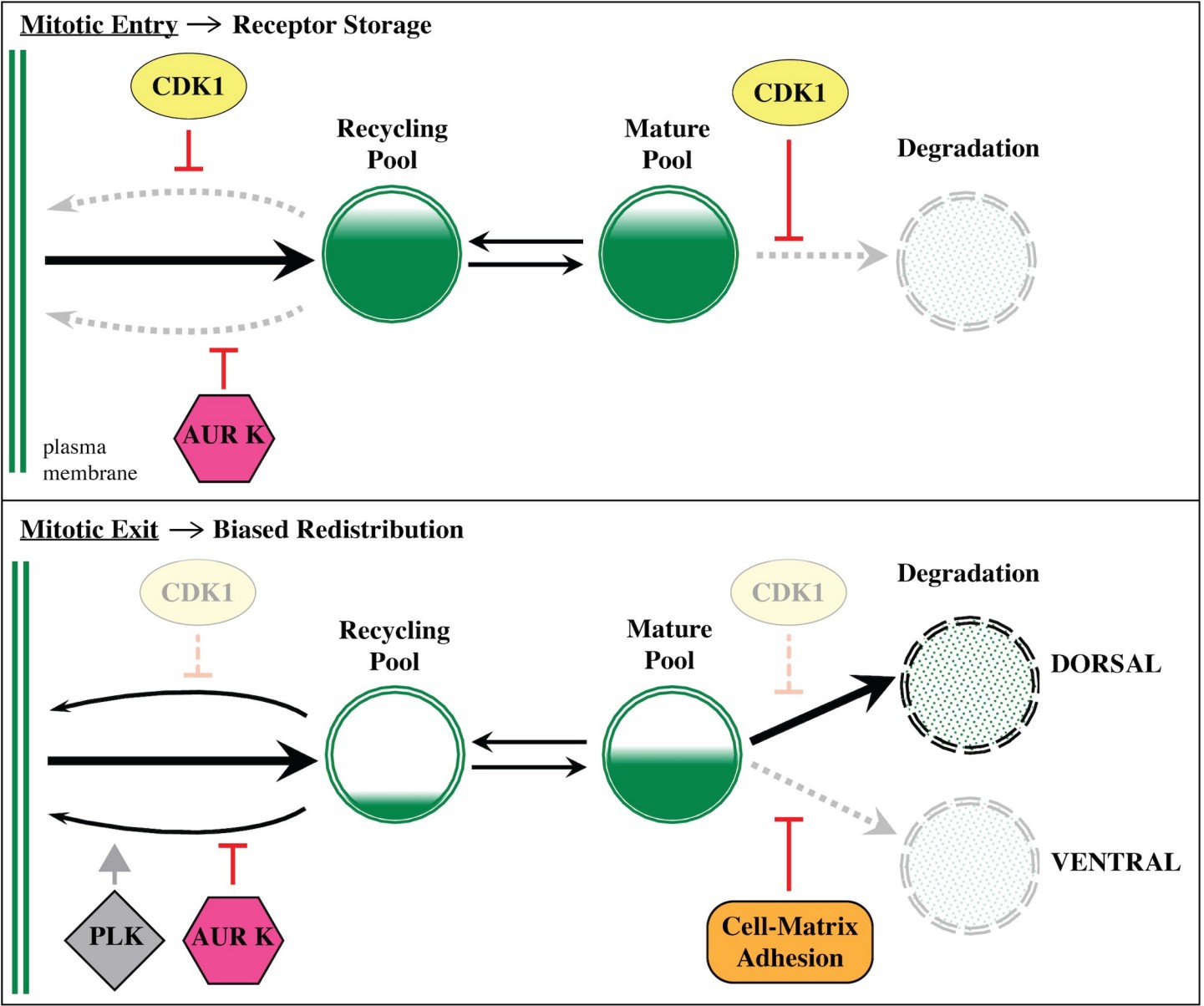

**Fig 6. Model for mitotic regulation of FGFR trafficking.** Diagrams illustrating hypothesized kinase-dependent shifts in trafficking and their impact on FGFR storage and redistribution during founder cell division. FGFR, Fibroblast Growth Factor Receptor.

## Methods

### Contact for reagent and resource sharing

Further information and requests for resources and reagents should be directed to and will be fulfilled by the Lead Contact, Brad Davidson (bdavids1@swarthmore.edu).

### Experimental model and subject details

*Ciona intestinalis* adults were collected and supplied by M-Rep (Carlsbad, California, United States of America) and maintained in the laboratory at 16 to 18 ˚C under constant illumination. Fertilization, dechorionation, and electroporation were carried out as previously described [40]. Embryos were staged according to [22].

## Method details

See S1 Table for a list of key Reagents/Resources used to generate the data presented in this study.

## Molecular cloning

The *Ci-Mesp* and *FoxF* enhancers were described previously [39,21]. *Mesp>LacZ, Mesp>FGFR::Venus*, and *Mesp>E-Cadherin::GFP* were previously described [20,21,41]. The CLIP and HALO open reading frames (ORFs) were PCR amplified from CLIP-rGBD Rho and HALO-rGBD Rho plasmids generously provided by William M. Bement using the primers: F ClipSnapNot/R Halo-CLIP Bam and inserted downstream of the *Mesp* enhancer using NotI and BlpI. To make *Mesp>Cyclin B$^{\Delta 90}$*, we PCR amplified *Ciona Cyclin B* from cDNA clone VES88_L15 using primers CyBDN Not1F and CyBDN EcoR1R to remove the sequence encoding the destruction box [25]. This fragment was swapped in place of *LacZ* in the *Mesp>LacZ* plasmid using the NotI and EcoRI sites. *Ciona* Rab4, Rab7, and Rab11 were PCR amplified using the following primer sets: Rab4_BamHI_F/ Rab4_BlpI_R, Rab7_BamHI_F/ Rab7_BlpI_R, and Rab11_BamHI_F/ Rab11_BlpI_R, from full ORF unigene collection (Cogenics) clones and inserted in frame using the BamHI and BlpI sites. The existing BlpI site was removed from Rab7 prior to amplification by site directed mutagenesis using Rab7noBlp_F/ Rab7noBlp_R primer set. *Mesp>HALO::Rab4$^{S199A/T200A}$*, *Mesp>HALO::Rab4$^{S199A}$*, and *Mesp>HALO::Rab4$^{S199D/T200D}$* were generated by site-directed mutagenesis of the *Mesp>HALO::Rab4* expression plasmid using following primer sets: Rab4ST_AA_F/Rab4ST_AA_R, Rab4S_A_F/Rab4S_A_R, and Rab4ST_DD_F/Rab4ST_DD_R. To make *Mesp>HALO::Vam2$^{421-841}$*, the region of *Ciona Vam2* corresponding to amino acids 421–841 was PCR amplified from a unigene collection clone using the following primers: VAM2_Forward/VAM2_Reverse and inserted downstream of HALO using the BamHI and BlpI sites. To make *Mesp>Cdki(p27)*, *Ciona Cdki(p27)* was PCR amplified from cDNA clone VES103_M15 using the primers: CKI_NotIF/CKI_BlpR and swapped in place of *LacZ* in the *Mesp>LacZ* plasmid using NotI and Blp1.

## Antibody staining/CLIP labeling

Embryos were fixed immediately after collection in approximately 2 mL of buffered paraformaldehyde (PFA) solution (4% PFA w/v, 0.1 M MOPS, 0.5 M NaCl, 0.1 mM EGTA, 2 mM MgSO$_4$ (pH 7)) in PBS overnight at 4˚C on a nutating mixer. Antibody staining was performed as previously described [20,40]. Briefly, embryos were washed 5 times in PBS-Triton X-100 (0.1% v/v), blocked with PBS-BSA (1% w/v) for 1 hour at room temperature and stained with 0.1% GFP Tag Monoclonal Antibody (3E6) in PBS-BSA overnight at 4˚C. Embryos were then washed 3 times in PBS-Triton, stained with 0.02% DRAQ5 in PBS-Triton for 1 hour at room temperature, washed 2 times in PBS-BSA, blocked with PBS-NDS (2% v/v) for 1 hour at room temperature, stained with 0.1% Alexa Fluor 488 donkey α-mouse antibody (Invitrogen A21202), Alexa Fluor Phalloidin 633 (to detect F-actin), and 0.5% CLIP-Cell TMR-Star in PBS-NDS for 2 hours at room temperature, washed 3 times in PBS-BSA, and mounted in approximately 75% glycerol.

## Inhibitor treatments

In order to inhibit CDK1 activity and induce prophase arrest in founder cells, *Ciona* embryos grown at 18˚C in filter sterilized sea water were treated with 5 μg/mL of Roscovitine approximately 10 minutes after blastopore closure (early Hotta Stage 14) and incubated for approximately 1 hour before fixation at Hotta Stage 16. In order to inhibit AurK activity and induce

prophase arrest in founder cells, *Ciona* embryos grown at 18 °C in filter sterilized sea water were treated with 10 μg/mL of VX-680 (Tozasertib) or 5 ug/mL of AMG-900 at Hotta Stage 13 and incubated for approximately 1.5 hour before fixation at Hotta Stage 16.

## Confocal microscopy and image processing

All images were acquired with a Leica SP5 confocal microscope (Leica Microsystems, Buffalo Grove, Illinois). For volumetric analysis, 12-bit z-stacks through the founder cells or TVC/ATM pairs were obtained through a 40× oil objective (N.A. 1.25) and 4× digital zoom with a step size of 0.3 μm. For live imaging, 12-bit z-stacks through the founder cells or TVC/ATM pairs were obtained through a 20× objective (N.A. 0.7) and 5× digital zoom with a step size of 1.0 μm. We scanned bidirectionally with a scan speed of 700 Hz and with cropping in the y-dimension to reduce imaging time. All images were recorded with 12-bit depth and the resolution set at $1024 \times 1024$. Image processing was performed using FIJI (ImageJ, N.I.H., Bethesda, Maryland) and Matlab (MathWorks, Natick, Massachusetts).

## Quantification and statistical analysis

**Cell segmentation.** Using FIJI (ImageJ) software, z-stacks were cropped to isolate individual founder cells for segmentation. Cell segmentation was performed using Matlab. Cropped images were smoothed using a 2D Gaussian filter and then binarized by thresholding. The threshold value was automatically calculated using Otsu's method and then scaled by the threshold level [42]. To fill in gaps, the cell mask was dilated, the holes were filled, and the cell mask was eroded in each z-plane. To remove regions outside of the cell, each mask was eroded in 3D, and objects with a volume of less than 10 $\mu m^3$ were deleted before the mask was redilated in 3D. This post-processing step was done to smooth the masks and to delete disconnected and minimally connected objects. Each cell mask was manually reviewed and, if necessary, the masks were adjusted for accuracy.

**Volumetric analysis.** Volumetric analysis was performed using Matlab. Images were smoothed using a 2D Gaussian filter with standard deviation of 0.1 μm, and then binarized by thresholding at the 95th percentile of pixel intensity within the cell mask. Thresholding was done to normalized the volume of the puncta to the volume of the cell mask, and the level of thresholding was selected based on the separation of signal from background across a set of sample of images taken from our dataset. To capture signal distribution and spatial colocalization, segmented cell volumes were divided into 3 regions based on the distance to the edge of the mask: plasma membrane-associated (0 to 1 μm), peripheral cytoplasm (1 to 3 μm), and deep cytoplasm (>3 μm). The 3D Euclidean distance was calculated with the linear time algorithm described by Maurer [43]. Importantly, the distance was adjusted to account for the voxel size of the 3D image. The FGF receptor fold enrichment was calculated for each region according to:

$$\text{Regional FGFR Enrichment} = \frac{V_{\text{FGF receptor in region}}/V_{\text{region}}}{V_{\text{FGF receptor in cell}}/V_{\text{cell}}}$$

where V = volume. The amount of signal in each region was normalized by volume to account for changes in cell morphology across images and mitotic stages. The resulting fold enrichment values were averaged and presented as mean ± standard error of the mean. Manders' Colocalization Coefficient [44] was calculated for whole cells and each region within these cells according to:

$$\text{Colocalization}_{cell} = \frac{V_{\text{FGF receptor} \cap \text{RAB in cell}}}{V_{\text{FGF receptor in cell}}}$$

$$\text{Colocalization}_{region} = \frac{V_{\text{FGF receptor} \cap \text{RAB in region}}}{V_{\text{FGF receptor in region}}}$$

where $V_{\text{FGF receptor} \cap \text{RAB in cell}}$ = the volume of FGFR::VENUS puncta that overlap with HALO:: RAB endosome puncta.

Founder cells display a strong adhesion-dependent cell polarity that results in ventrally biased FGFR distribution [20,41]. To determine whether ventral FGFR polarization impacted the results of our volumetric analysis, total cell volume for each segmented founder cell was divided in half along the dorsal–ventral axis. Analysis of the mitotic FGFR distribution in ventral regions of our founder cells mirrored the results from whole cell analysis. These results indicate that stage-specific shifts in FGFR distribution primarily reflect changes on the ventral side of polarized founder cells. consistent with previous data [20]. We also used these Ventral/ Dorsal volumes to calculate Ventral/Dorsal enrichment ratios,

$$\text{Ventral/Dorsal FGFR Enrichment} = \frac{V_{\text{FGF receptor in ventral region}}}{V_{\text{FGF receptor in dorsal region}}}$$

## Statistical analysis

In all graphs, error bars represent standard error of mean (SEM) as stated in the results and figure legends. Statistical significance was determined using one-way ANOVA followed by Tukey multiple comparison test unless otherwise indicated in the results or figure legends.

## Supporting information

**S1 Fig. Inhibition of mitotic entry suppresses FGFR mitotic trafficking but does not impact TVC induction (related to Fig 1).** (**A-B'**) Ventral projections and lateral sections for founder cells electroporated as indicated. Dashed lines (**A and B**; orange) indicate position of sections (**A' and B'**). (**C**) Graphical summary of regional FGFR::VENUS enrichment for founder cells electroporated as indicated. No significant changes in regional FGFR::VENUS enrichment were detected in arrested *Mesp>Cdki(p27)* transgenic founder cells (plasma membrane-associated $p = 0.489$, peripheral cytoplasm $p = 0.527$, deep cytoplasm $p = 0.899$). Data were obtained from 2 independent trials, $n > 16$. (**D**) Graphical summary of mitotic arrest at different stages as observed for founder cells electroporated with either *Mesp>LacZ* or *Mesp>Cdk1(p27)* as indicated. Data were obtained from 3 independent trials, $n > 13$ per trial. (**E-F"**) Representative micrographs of late tailbud embryos showing cranial-cardiac progenitor induction (indicated by overlapping *Mesp>Ensc*::GFP and *FoxF>RFP* reporter expression) versus noninduced precardiac founder lineage cells (indicated by *Mesp>Ensc*::GFP reporter expression alone) in embryos coelectroporated with either *Mesp>LacZ* or *Mesp>Cdk1(p27)* as indicated [20,40,41,21]. (**G-H**) Graphical summary of mitotic arrest and heart progenitor induction in embryos cotransfected as indicated. Data were obtained from 3 independent trials, $n > 17$ per trial. Scale bars are indicated in micrometers. Significance indicated; n.s., not significant. Significance was determined using one-way ANOVA followed by Tukey multiple comparison test. Error bars represent SEM. Numerical values for all graphs can be found in S6 Data. ATM, Anterior Tail Muscle Cell; FGFR, Fibroblast Growth Factor Receptor; SEM, standard error of mean; TVC, Trunk ventral cell/Cranial-cardiac progenitor. (PDF)

**S2 Fig. Stage-specific quantitation of mitotic FGFR trafficking patterns (related to Fig 2).** (**A-A'''**) Graphical summary of whole cell (**A**) and regional FGFR::VENUS/ CLIP::RAB7

colocalization (**A'-A'''**; Manders' overlap) during founder cell division (**data shown correspond to data presented in Fig 2D**). $n > 6$ for each mitotic stage. Regional overlap was measured in 3 concentric regions, plasma membrane, peripheral cytoplasm, and deep cytoplasm (**Fig 1A-A''**; Methods). Lack of any significant change ($p > 0.05$) is indicated by no change in lettering (a for all columns). Significance was determined using one-way ANOVA followed by Tukey multiple comparison test. Numerical values for all graphs can be found in S7 Data. Error bars represent SEM. FGFR, Fibroblast Growth Factor Receptor; SEM, standard error of mean.
(PDF)

**S3 Fig. Inhibition of CDK1 does not impact endosomal maturation or slow recycling of FGF receptors during mitotic entry (related to Figs 2 and 3).** (**A-B'**) Masked/thresholded transverse sections of founder cells electroporated with *Mesp>FGFR::Venus* and *Mesp>HALO::RAB11* and treated as indicated. For clarity, panels showing only colocalized FGFR::VENUS/CLIP::RAB11 puncta are provided (OVERLAP; Manders' overlap; MOC) (**A'** and **B'**). (**C-E**) Graphical summary of whole cell (**C**) and regional FGFR::VENUS/ CLIP:: RAB11 colocalization (**D-E**; Manders' overlap) in founder cells treated as indicated. (**F-H**) Graphical summary of whole cell (**F**) and regional FGFR::VENUS/ CLIP::RAB4 colocalization (**G-H**; Manders' overlap) in founder cells treated as indicated. Data were obtained from 2 independent trials, $n > 14$. Scale bars are indicated in micrometers. Significance indicated by *p*-value or a change in lettering (a versus b). Lack of significance indicated by n.s. Significance was determined using one-way ANOVA followed by Tukey multiple comparison test. Numerical values for all graphs can be found in S8 Data. CDK1, Cyclin-dependent Kinase 1; FGF, Fibroblast Growth Factor; n.s., not significant.
(PDF)

**S4 Fig. Prolongation of CDK1 activity leads to excessive FGFR internalization and blocks TVC induction (related to Fig 2).** (**A-B'**) Ventral projections and lateral sections for founder cells electroporated as indicated. Dashed lines (**A and B**; orange) indicate position of sections (**A' and B'**). (**C**) Graphical summary of regional FGFR::VENUS enrichment for founder cells electroporated as indicated (deep cytoplasm; $p = 0.264$). Data were obtained from 2 independent trials, $n > 7$. (**D**) Graphical summary of mitotic arrest observed for founder cells electroporated. Data were obtained from 3 independent trials, $n > 22$ per trial. (**E-F''**) Representative micrographs of late tailbud embryos showing cranial-cardiac progenitor induction (indicated by overlap of *Mesp> Ensc::GFP* and *FoxF>RFP* reporter expression along with migration into the head/trunk region) versus noninduced precardiac founder lineage cells (indicated by *Mesp>Ensc::GFP* reporter expression alone along with lack of migration) in embryos coelectroporated with either *Mesp>LacZ* or *Mesp>CyclinB$^{A90}$* as indicated [20,40,41,21]. Note that prolongation of CDK1 activity appears to disrupt induction. This may be due to failure of transgenic cells to properly exit mitosis or it may reflect observed FGFR internalization. (**G-H**) Graphical summary of mitotic arrest and heart progenitor induction in embryos cotransfected as indicated. Data were obtained from 3 independent trials, $n > 8$ per trial. Arrested *Mesp>CyclinB$^{A90}$* transgenic embryos (**A-C**) were fixed and analyzed at Hotta Stage 16 [22], approximately 1 hour after control cells (*Mesp>LacZ*) complete asymmetric division. Scale bars are indicated in micrometers. Significance was determined using one-way ANOVA followed by Tukey multiple comparison test. Numerical values for all graphs can be found in S9 Data. Error bars represent SEM. ATM, Anterior Tail Muscle Cell; CDK1, Cyclin-dependent Kinase 1; FGFR, Fibroblast Growth Factor Receptor; SEM, standard error of mean; TVC, Trunk ventral cell/Cranial-cardiac progenitor.
(PDF)

**S5 Fig. Inhibition of both CDK1 Kinase activity and lysosomal degradation increases the plasma membrane-associated enrichment of FGF receptors.** (**A-B'**) Lateral sections and graphical summary of regional FGFR::VENUS enrichment for founder cells electroporated with *Mesp>FGFR::Venus* alone or in combination with *Mesp>HALO::Vam2$^{421-841}$* and treated with vehicle (DMSO) or Roscovitine (14 μmol/L) as indicated. *Mesp>HALO::Vam2$^{421-841}$* alone also resulted in a modest, but not significant, increase in plasma membrane-associated FGFR::VENUS. Because phalloidin staining obscures FGFR::VENUS localization, red dashed lines were used to indicate phalloidin-stained cell membranes (**A-B**). Some regions are labeled with an a or b to denote significant changes ($p < 0.05$) that occurred within this region across stages. Other regions are labeled n.s. to denote that no significant changes occurred for the indicated stages. Significance was determined using one-way ANOVA followed by Tukey multiple comparison test. (**C**) Quantification of the FGFR::VENUS enrichment in the plasma membrane-associated region of founder cells electroporated and treated as indicated. Significance was determined using one-way ANOVA followed by Tukey multiple comparison test. Numerical values for all graphs can be found in S10 Data. Scale bars are indicated in micrometers. CDK1, Cyclin-dependent Kinase 1; FGF, Fibroblast Growth Factor.
(PDF)

**S6 Fig. RAB4 phosphomutants impact TVC induction (related to Fig 4).** (**A-D"**) Representative micrographs of late tailbud embryos showing induced cranial-cardiac progenitors (TVCs, arrowheads point to cells showing overlapping *Mesp>GFP* and *FoxF>RFP* reporter expression) versus noninduced anterior muscle lineage cells (ATMs, arrows point to cells showing *Mesp>GFP* reporter expression alone) in embryos coelectroporated with *Mesp>LacZ* ($n = 258$), *HALO::Rab4* ($n = 235$), *HALO::Rab4$^{S199A/T200A}$* ($n = 277$), or *HALO::Rab4$^{S199D/T200D}$* ($n = 130$) as indicated [20,40,41,21]. (**E**) Graphical summary of heart progenitor induction in embryos cotransfected as indicated. Embryos electroporated with *HALO::Rab4$^{S199A/T200A}$* show increased induction as indicated by the increased proportion of cells with overlapping *Mesp>Ensc::GFP* and *FoxF>RFP* in comparison to control embryos electroporated with Mesp>*LacZ* ($p = 0.02$) or *HALO::Rab4* ($p = 0.02$). Embryos electroporated with *HALO::Rab4$^{S199D/T200D}$* show decreased induction as indicated by the increased proportion of cells with *Mesp>Ensc::GFP* but no *FoxF>RFP* in comparison to control embryos electroporated with *Mesp>LacZ* ($p = 0.0001$) or *HALO::Rab4* ($p = 0.006$). Data were obtained from 3 independent trials, $n > 31$ per trial. Scale bars are indicated in micrometers. Significance was determined using a *t* test with an arcsine square root transformation. Numerical values for all graphs can be found in S11 Data. Error bars represent SEM. ATM, Anterior Tail Muscle Cell; SEM, standard error of mean; TVC, Trunk ventral cell/Cranial-cardiac progenitor.
(PDF)

**S7 Fig. Inhibition of Aurora Kinase activity does not impact fast recycling of FGF receptors during mitotic entry or RAB7 or RAB11 overlap in the deep cytoplasm (related to Fig 4).** (**A-C**) Graphical summary and quantitative analysis of regional FGFR::VENUS enrichment for founder cells electroporated with *Mesp>FGFR::Venus* and treated with vehicle (DMSO) or AMG-900 (10 μmol/L) as indicated. (**D**) Quantification of regional FGFR::VENUS/CLIP:: RAB11 overlap in founder cells electroporated and treated as indicated. (**E-G**) Masked/thresholded transverse sections and quantification of regional FGFR::VENUS/CLIP::RAB4 overlap for founder cells electroporated and treated as indicated. MOCs for whole cell analysis are indicated (**E and F**) Note that treatment with VX-680 had no significant impact on Rab4 colocalization (**E-G**). Treatment with AMG-900 also had no significant impact [whole cell overlap for DMSO-treated cells MOC = $0.155 \pm 0.019$ ($n = 7$) and AMG-900 treated cells MOC = $0.112 \pm 0.015$ ($n = 3$) $p = 0.118$]. (**H-K**) Graphical summary and quantification of

regional FGFR::VENUS/CLIP::RAB7 overlap in founder cells electroporated and treated as indicated. (**L-O**) Graphical summary and quantification of regional FGFR::VENUS/CLIP:: RAB7 overlap in founder cells electroporated and treated as indicated. Data were obtained from 2 independent trials. $n$ = number of founder cells analyzed. Scale bars are indicated in micrometers. Significance indicated by asterisk and/or change in letter. n.s., not significant. Significance was determined using one-way ANOVA followed by Tukey multiple comparison test. Numerical values for all graphs can be found in S12 Data. FGF, Fibroblast Growth Factor; MOC, Manders' overlap coefficient.
(PDF)

**S1 Table. Key Reagents/Resources used to generate the data presented in this study.**
(PDF)

**S1 Data. The raw data associated with all graphs found in Fig 1.**
(XLSX)

**S2 Data. The raw data associated with all graphs found in Fig 2.**
(XLSX)

**S3 Data. The raw data associated with all graphs found in Fig 3.**
(XLSX)

**S4 Data. The raw data associated with all graphs found in Fig 4.**
(XLSX)

**S5 Data. The raw data associated with all graphs found in Fig 5.**
(XLSX)

**S6 Data. The raw data associated with all graphs found in S1 Fig.**
(XLSX)

**S7 Data. The raw data associated with all graphs found in S2 Fig.**
(XLSX)

**S8 Data. The raw data associated with all graphs found in S3 Fig.**
(XLSX)

**S9 Data. The raw data associated with all graphs found in S4 Fig.**
(XLSX)

**S10 Data. The raw data associated with all graphs found in S5 Fig.**
(XLSX)

**S11 Data. The raw data associated with all graphs found in S6 Fig.**
(XLSX)

**S12 Data. The raw data associated with all graphs found in S7 Fig.**
(XLSX)

## Acknowledgments

We thank Danelle Davenport for her suggestions and comments on this study. We also thank William (Bill) Bement for his generous gift of HALO and CLIP constructs. We thank Dong Shin (Chris) You for his work in establishing the FGFR segmentation protocol. We thank Johnathan White for his work in establishing the live imaging protocol.

## Author Contributions

**Conceptualization:** Christina D. Cota, Brad Davidson.

**Data curation:** William Colgan.

**Formal analysis:** Christina D. Cota, William Colgan, Twan Sia.

**Funding acquisition:** Brad Davidson.

**Investigation:** Christina D. Cota, Matthew S. Dreier, Anna Cha, Twan Sia.

**Methodology:** Christina D. Cota, Brad Davidson.

**Project administration:** Brad Davidson.

**Resources:** Brad Davidson.

**Software:** William Colgan.

**Supervision:** Christina D. Cota, Brad Davidson.

**Validation:** Christina D. Cota, Brad Davidson.

**Visualization:** Christina D. Cota, Brad Davidson.

**Writing – original draft:** Christina D. Cota, Matthew S. Dreier, William Colgan, Twan Sia, Brad Davidson.

**Writing – review & editing:** Christina D. Cota, Twan Sia, Brad Davidson.

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
