## [Editor Report · Decision Letter 0]

4 Feb 2020

Dear Dr Davidson, 

Thank you for submitting your manuscript entitled "Mitotic Kinases choreograph receptor storage and redistribution." for consideration as a Research Article by PLOS Biology.

Many apologies for the delay in getting back to you and thank you for your patience when we assessed your submission. Your manuscript has now been evaluated by the PLOS Biology editorial staff as well as by an academic editor with relevant expertise and I am writing to let you know that we would like to send your submission out for external peer review.

Please note, however, that the outcome of our discussion of your manuscript is that we have some reservations as to the depth of analysis. We would need to be persuaded by the reviewers that the paper has the potential after revision to offer the significant strength of advance and sufficient experimental support that we require for publication in order to pursue it further for PLOS Biology. Presently, we are concerned about the over-reliance on the localisation data to draw conclusions and a lack of direct evidence on CDK1-dependent RAB4 phosphorylation. We are letting you know now as you may wish to begin work on experiments that can address these concerns now. There is, however, absolutely no guarantee of the outcome of the review process.

Before we can send your manuscript to reviewers, we need you to complete your submission by providing the metadata that is required for full assessment. To this end, please login to Editorial Manager where you will find the paper in the 'Submissions Needing Revisions' folder on your homepage. Please click 'Revise Submission' from the Action Links and complete all additional questions in the submission questionnaire.

Please re-submit your manuscript within two working days, i.e. by Feb 06 2020 11:59PM.

Kind regards,

Di Jiang

PLOS Biology

---

## [Decision Letter · Decision Letter 1]

25 Feb 2020

Dear Dr Davidson,

Thank you very much for submitting your manuscript "Mitotic Kinases choreograph receptor storage and redistribution." for consideration as a Research Article at PLOS Biology. Your manuscript has been evaluated by the PLOS Biology editors, an Academic Editor with relevant expertise, and by four independent reviewers.

In light of the reviews (below), we will welcome re-submission of a revised version that takes into account the reviewers' comments, in particular use of alternate inhibitors. We cannot make any decision about publication until we have seen the revised manuscript and your response to the reviewers' comments. Your revised manuscript is also likely to be sent for further evaluation by the reviewers.

We expect to receive your revised manuscript within 2 months. 

**IMPORTANT - SUBMITTING YOUR REVISION**

*Re-submission Checklist*

*Published Peer Review*

*PLOS Data Policy*

*Blot and Gel Data Policy*

Sincerely,

Di Jiang

PLOS Biology

REVIEWS:

Reviewer #1: As someone who has worked in mitosis, polarity and membrane trafficking, this manuscript is potentially of great interest to cell and developmental biology. However, the major weakness is that the manuscript is incredibly hard to understand and the results are not always clearly presented. So, I fear that the non-specialist will not be able to follow it without substantial improvements to the clarity of presentation. 

Comments:

1. To make their findings more understandable to the non-specialist, the authors could consider converting part of the current Figure S1 into a revised Fig 1, to show where these founder cells are located within the embryo, and to show the cross-section of the wild type FGFR::Venus at interphase, metaphase and telophase, which is currently only schematized in Fig 1A. The authors should not assume that the reader has read and can recall their previous papers. Rather, they should feel comfortable in simply showing new versions of previously demonstrated data or schematics again, at least of the wild-type cells and what happens to them during mitosis, so that the basic phenomenon is clearly established in Fig 1 without needing to read the text or previous papers. For example, why does the previous Dev Cell paper show a schematic of a the Ciona embryo founder cells and their location within the embryo in Fig 1A, while this new paper does not? It would also be worth showing new examples of the Fig 1B-D in the Dev Cell paper in this new paper. In particular, there is no interphase control cell shown for Fig 1B.

2. In general, all of the microscopy images of the founder cells are too small to easily see what is going on. It would be best to increase the size of each of the panels by around 400% at the expense of the white space currently occupying most of the figures. Otherwise, it is hard to see any substantive differences between any of the panels, or to see any co-localisation.

3. In the previous Dev Cell paper, the FGFR appears to move completely (100%) from the lateral membranes at interphase to endosomes in mitosis (and is asymmetrically distributed). In this paper, the authors quantify this change but only observe a 30% reduction in plasma membrane-associated FGFR (Fig 1D). What is the reason for this difference?

4. I find the Roscovitine (CDK1 inh.) experiment deeply confusing. If mitotic entry is blocked, shouldn't the FGFR stay at the plasma membrane? The fact that it doesn't (and is degraded) is interpreted as these cells being 'blocked in prophase'. But what is the evidence for this? Secondly, if active CDK1 is needed to inhibit degradation of FGFR, then why isn't FGFR degraded during interphase, when CDK1 is not active? I would suggest repeating the Roscovitine experiment in interphase cells - is the FGFR also degraded there? The authors must already have done this control. Also, is it clear that VX680 doesn't also lead to inhibition of CDK1 through loss of the positive feedback loop between AuroraA/B and CDK1 activation? If so, then perhaps the Roscovitine is causing an artefactual result, which the authors should consider seriously before pressing ahead with publication.

Reviewer #2 (Emmanuel Boucrot, signed review): Cota, Davidson and colleagues are reporting that mitotic kinases Cdk1 and Aurora are suppressing endosomal degradation and recycling back to the plasma membrane of Fibroblast Growth Factor Receptors (FGFRs) during asymmetric division. 

The authors followed-up on their seminal 2015 paper in which they reported that FGFR are internalised into endosomes during pre-cardiac founder cell division in the chordate Ciona robusta. This leads to an unequal distribution of FGFRs between the two daughter cells, thereby causing differential cell fate, which is critical for proper development. The study of mitotic FGFR distribution in Ciona founder cells is a powerful model of highly physiological relevance. 

In the present study, the authors measured with precision FGFR distribution between daughter cells and localisation inside late, fast- and slow-recycling endosomes (Rab7-, Rab4- and Rab11-positive compartment, respectively). They found that the receptors are stored into Rab11- and Rab7-endosomes during early mitosis and recycled back to the plasma membrane during mitotic exit. They further established that the mitotic kinase CyclinB1/CDK1 blocks the Rab4-dependent fast recycling route whereas Aurora controls the fusion back to the plasma membrane from Rab11-positive compartments. They also establish that both kinases controls late endosome to lysosome conversion, tipping the balance towards the formers and thereby sparing FGFRs from degradation.

The work was carefully planned, executed and analysed with the appropriated controls and statistics to support most of the conclusions. The findings are new and deepen significantly our understanding of membrane trafficking during mitosis and endosome inheritance upon cell division. 

The only weak part of the manuscript is the section on Aurora kinase (Figure 5): 

1. Unlike for CDK1 where the authors backed up their results with small compound inhibitor with a genetic approach (the CyclinBDelta90 construct), all the conclusions on Aurora are based on the inhibitor VX-680. However, as most small compound inhibitor, VX-680 is not strictly specific and also inhibits Src, Abl, GSK3beta as well as few other kinases (see PMID 16424036). Its effect on GSK3beta is potentially relevant, as explained below. 

Thus, it would be prudent to confirm the phenotype using a second, chemically unrelated, inhibitor such as AMG-900 (PMID 20935223) that do not share the same secondary targets (p38alpha, DDR1 and 2 and LTK for AMG-900). Alternatively, if possible in Ciona, a genetic approach (depletion, dominant negative or constitutively active constructs) would alleviate concerns about small compound inhibitors.

2. Contrasted with CDK1 where the authors could confirm that the phenotype is mediated by the phosphorylation of S199 on Rab4, the lack of molecular insight about how Aurora works makes for a weak ending to the paper. 

The VX-680 phenotype (Fig. 5A-C) is consistent with a block of Rab11-positive recycling endosome back to the cell surface. This could be a defect in RE transport within the cytoplasm or an inhibition of RE to plasma membrane (PM) fusion. Both of these steps are controlled by Rab11 through its interaction with the exocyst complex (Exo70 etc.), Arf6 and FIP3, 4 and 5 proteins. 

Of potential relevance, FIP5 was reported to be phosphorylated on T276 by GSK3beta during mitosis (PMID 24591568). Interestingly, the motif LLT276*RS read from right to left (kinases obviously read amino acids in both direction) is SRT*LL which could be an Aurora B consensus site [R/K][S/T][I/L/V]. With the dual inhibition of Aurora and GSK3beta by VX-680, it is pressing to rule out any role for GSK3beta (using the CHIR-99021 or BIO inhibitors or genetic perturbations such as knock-down or S9A- or K85A- GSK3beta constitutively active and kinase dead mutants, respectively). 

It is also possible that Aurora kinase acts on the SNAREs, which in the case of recycling endosomes are VAMP2 or 3, SNAP23 and Syntaxin 4. Inhibiting VAMP3 function perturbed TfR-containing recycling endosome fusion back to the cell surface during late mitosis (PMID 17483462) and there are links between Aurora and SNARE function in late cytokinesis (PMID 16213214; 19887622). 

I appreciate that a detailed study of how Aurora blocks RE fusion to the PM is beyond the scope of this manuscript but addressing point 1 (required for publication in PLOS Biology in my opinion) and providing some insights into point 2 would greatly benefit to the impact and robustness of the paper. 

Minor comments:

- the paragraph title "Endocytic pathways involved in mitotic redistribution of FGFR" (page 4) is misleading as one would expect a study of the pathways by which the receptor enter cells during mitosis (i.e. clathrin-mediated endocytosis, CLIC/GEEC, FEME, macropinocytosis etc..). As the authors studied the endosomal paths used by the receptor (fast- and slow-recycling or degradation), "Endosomal pathways involved in […]" would be more appropriated. 

- Page 6 (top) and page 7 (top) "roscovatine" should be "roscovitine" 

Reviewer #3: This paper from Cota and colleagues sheds light on the regulation of endosomal trafficking and receptor sorting during asymmetric cell division.

Endosomal trafficking recently emerged as a major regulator of morphogenesis thought its effects on signalling, both for symmetric cell division (to ensure equal transmission of morphogen molecules between daughter cells) and for asymmetric cell division (to ensure asymmetric cell fate). However, while decades of cell biology provided a deep understanding of the regulation of trafficking during interphase in cultured cells, very little is known about the regulation of trafficking during mitosis in tissues. This is mostly due to the fact that tissues are less amenable to quantitative imaging studies than cultured cells, but also due to the fact that until recently, endocytic uptake was thought to be shut down during mitosis. This quantitative study addresses this question directly, in the physiological context of asymmetric cell division in the chordate Ciona robusta.

The authors first show using quantitative 3D imaging that FGFR receptors are internalized during early mitosis, before going back to the plasma membrane at the end of mitosis. They then show that internalized receptors do colocalize with recycling endosomes (Rab4-Rab11) and late endosomes (Rab7). They then use drugs to characterize the effect of mitotic kinases (Aurora A and CDK1) on this FGFR trafficking. They then provide evidences that CDK1 inhibits FGFR degradation in lysosomes and inhibit fast recycling through the Rab4 pathway. Last, they provide evidences that Aurora kinase inhibits Rab11-dependant slow recycling, as well as increase receptor trafficking to late, Rab7 compartments.

I think this is an excellent paper addressing a fundamental question of trafficking in a physiologically relevant context. I particularly appreciated the efforts the authors put in being very quantitative. I found particularly fascinating the finding that CDK1 specifically protects from degradation only a subset of receptors (like FGFR) rather that uniformly inhibiting the degradation pathway. This has important implications for our molecular understanding of morphogenesis, in particular because tissue-scale variations of receptor degradation in lysosomes is thought to control the establishment of morphogen gradients. While I think the study deserves few controls to ensure the specificity of the treatments used (see below), I think this paper fully deserves publication in PloS biology. 

Specific comments:

Main comment: specificity of the CDK1/Aurora treatment

The first part of the study deals with effects ascribed to the CDK1 kinase using the inhibitor Roscovitine. It is my understanding that this inhibitor is broad and can target other CDK, including CDK2 and CDK5 as well as other kinases (ERK2), albeit with less affinity (Meijer et al, Eur J Biochem. 1997). To address this problem, the authors nicely made use of the CyclinBDelta90 overexpression, which shows increased FGFR signals, in good agreement with their Roscovitine data (Fig 3I). But this panel i) does not have a control (meaning a live FGFR::VENUS expressing cells, showing the near absence of signal) and ii) is not quantified (it should be easy enough to quantify as the authors did in Fig 3E).

I think this part of the work deserves further characterization to ensure that the effects are indeed CDK1-specific. First, by quantifying the CyclinBDelta90 effect, but also by using other CDK1 inhibitors with a different spectrum like BMS-265246 or R547 (there are probably others). Obviously, it is probably not worth redoing ALL the Roscovitine experiments with another inhibitor, but at least Figure3 A,B-D would be nice.

Similarly, the last figure deals with Aurora inhibition and all data solely comes from one inhibitor, VX-680. This inhibitor inhibits Aurora in the nM range in vitro, which is orders of magnitude below the concentration used by the authors (20uM). This increases the chances of off-targets, which have been reported for this molecule (Cheetham GM, et al. Cancer Lett. 2007)(obviously we do not know if it's true in Ciona as well, perhaps the authors do). I think it would strengthen the paper to confirm these results with either another inhibitor, or a dominant active/inactive Aurora for instance.

Minor comments:

-3D segmentation. I really appreciated the care shown by the authors to use unbiased, user-free thresholding methods. However, I did not understand how the authors segmented the plasma membrane from the VGFR staining (i.e. I don't understand how they come up with the white perimeter of the cell from the green segmented region in Fig 1B'). Please explain in a bit more details.

-I really like the representation and colour coding scheme chosen by the authors in Fig1C. I think it would be nice to add a snapshot of a cell for each mitosis phase to appreciate the raw data along the cell cycle (in supplementary material at least). 

In addition, I think the "a, b, ns" in Fig 1C and D-D'' (and the rest of the paper) are a bit confusing. Since the authors used different colours when differences are statistically significant with the previous phase, why not simply stating it in the legend? The graph makes the point anyway. 

- The authors are excellent at testing the significance relevance of nearly every observation, but I could not find the number of cells analysed for each condition in Fig 1D-D'', Fig 2 and Fig 4C. 

-Fig3K: Why is there DMSO in the control if there is no drug in the Vam2 treatment? 

-Fig3 A',B',C',G' it seems that there is some Rab7 signal (magenta) within the nucleus (Cyan). Why is that ?

-Fig 4C. I think a control with HALO::Vam2[421-841] without Roscovitine would be useful (it will probably enhance the effect).

-The inhibitor treatment section of the methods refers to a Brefeldin A treatment, but I could not find any reference to it in the text or the figures. Also, I think that 10ug/mL of VX680 corresponds to 21uM not 10uM as stated in the legends of Fig 3 and 5 (MW 464.59). 

Reviewer #4: Review Cota et al

General comments

As cells enter mitosis and round-up the total amount of plasma membrane is reduced through a process that prevents recycling of endosomes back to the plasm membrane. Recycling occurs either via molecularly distinct fast or slow pathways. How these two recycling pathways are shut off during mitosis is currently unknown. 

Here the authors nicely demonstrate that the major kinase that drives cells into mitosis (CDK1) prevents fast endosome recycling. In addition, the authors demonstrate that CDK1 also prevents lysosomal degradation of internalized FGF receptors, leading to their accumulation within the cell. To supplement this analysis the authors also show that Aurora kinase (another kinase active in mitosis) prevents the slow recycling pathway. These two kinases thus lead to the accumulation of endocytosed FGF receptors on internal endosomes, allowing them to be recycled to the plasma membrane once the cell exits mitosis. Intriguingly, it is also suggested that the subcellular distribution of internalized FGF receptors is controlled by Aurora kinase, and that this has implications for the cell fate outcome of the two daughter cells (either cranial-cardiac or tail muscle: FoxF is used as an indicator of heart progenitor fate).

This is a well-written article that addresses a fundamental topic using both pharmacological and molecular tools combined with sophisticated volumetric image analysis of fluorescent reporter constructs targeted via transgenesis to the specific cells. I support publication of this article following some minor revision.

One general comment is related to the previous article published by the group (Cota and Davidson, 2015), where they nicely demonstrated that FGF receptors were degraded in mitosis (reversed by MG132 mitotic block) when cell adhesion was inhibited with RapS17N. I therefore expected an update on how lack of adhesion leads to receptor degradation during mitosis given the current finding that CDK1 also shuts off the FGF receptor degradation machinery during mitosis.

Figures. 

Figure 1. FGF receptor moved from plasma membrane to internal stores then back to plasma during mitotic entry, mitosis and mitotic exit respectively.

This figure was fine.

Figure 2. Dynamics of FGFR cycling and association with RAB7 or RAB11 endosomes

Late endosome RAB7, slow recycling endosome RAB11 - OK

Prophase: FGFR and RAB11 peripheral co-localization increased = Slow Cycling Endosomes - OK

Metaphase: FGFR and RAB7 co-localization increased (deep cell Maturation Pathway), FGFR and RAB11 co-localization remained unchanged - OK

Anaphase: FGFR and RAB7 co-localization decreased (recycled or degraded), FGFR and RAB11 peripheral co-localization increased (and returned to plasma membrane) - OK

Figure 3. CDK1 suppresses FGFR degradation during mitotic entry

3A,B. Blocking cells entering mitosis with Roscovitine led to decrease in FGFR staining.

3C. Blocking cells entering mitosis with VX-680 (Aurora kinase inhibitor) did not reduce FGFR staining.

3F-H. E-Cadherin staining not affected by Roscovitine.

Conclusion: CDK1 stabilizes a subset of membrane proteins

Hypothesis: CDK1 suppresses FGFR lysosomal degradation.

3D. Blocking lysosomal degradation with Vam2 expression rescued FGFR signal loss induced by Roscovitane - Nice

3I and Figure S4. A stabilized form of cyclin B (delta 90) that causes sustained CDK1 activity increased FGFR staining - consistent and opposite to Roscovitine finding. 

However, the quantification of the delta 90 experiment is missing. Please add.

Conclusion: CDK1 activity suppresses RAB4 fast recycling and promotes accumulation of internalized FGFR by also preventing FGFR destruction.

Thus, inhibiting CDK 1 activity should lead to more plasma membrane or peripheral FGFR if FGFR destruction is also blocked (with for example MG132).

Authors were unable to test this because inhibiting CDK1 also prevents FGFR destruction

However, the authors could have combined Roscovitine with MG132 to test this prediction. This experiment may be worth doing of possible.

Figure 4. CDK1 phosphorylation of RAB4 suppresses cycling

Phosphorylation site on RAB4 conserved - interesting that the proline is not conserved.

Phospho-mimetic RAB4 (inactive mutant S199/A199) drove enrichment of FGFR on PM and reduction in peripheral cytoplasm during prophase - nice.

Phospho-mimetic RAB4 (active mutant S199/D199) blocked FGFR recycling during anaphase - nice.

Conclusion: CDK1 phosphorylation of RAB4 inhibits fast receptor recycling during mitosis.

Note - in the cartoon it would appear that blocking Rab4 phorphorylation (S199A) should lead to accumulation of Rab4 vesicles in the periphery on slow recycling vesicles. However, the data indicates that Rab4 is higher on the plasma membrane. Please provide cartoon that displays the data provided in Figures 4H and K. Also, I was not clear why there was not more Rab4 recycled to the plasma membrane via the fast recycling pathway. 

Figure 5. Aurora kinase regulates maturation and recycling of FGFR endosomes.

VX-680 increased FGFR on plasma membrane - OK.

Hypothesis: Aurora kinase blocks exocytosis of RAB11/FGFR endosomes during mitotic entry.

Aurora kinas inhibition blocked co-localization of RAB11 and FGFR.

Interestingly, VX-680 also decreased RAB7/FGFR co-localization in plasma membrane/peripheral regions.

This result suggests that Aurora kinase promotes maturation of FGFR enriched endosomes.

Hypothesis: this late endosomal pool is important for ventral polarization via biased degradation on founder cell's dorsal side. 

In line with this hypothesis, VX-680 gave increased FGFR on founder cell's dorsal side.

Figure 6. Model

OK, although large solid arrows appear unchanged throughout.

Suggestion

Is this specific for FGFR. Could test with Calveolin as previously, for example with D90 does Calveolin behave the same as FGFR?

Discussion

Good, although I was left wondering about the relative contributions of adhesion-dependent suppression of FGFRs during mitosis and CDK1-dependent suppression of FGFRs during mitosis, and also how loss of adhesion bypasses the CDK1 suppression of FGFR degradation.

Minor comments

Page 3, Line 3 … for promoting exit specific… Should Be …for promoting exit from specific…

Fig. S5A. Two scale bars appear on same image.

Roscovitine or Roscovatine. Please change all to Roscovitine in the text.

---

## [Decision Letter · Decision Letter 2]

11 Nov 2020

Dear Dr Davidson,

Thank you for submitting your revised Research Article entitled "Mitotic Kinases choreograph receptor storage and redistribution." for publication in PLOS Biology. I have now obtained advice from three of the original reviewers and have discussed their comments with the Academic Editor. 

Based on the reviews, we will probably accept this manuscript for publication, assuming that you will modify the manuscript to address the remaining points raised by the reviewers. Please also make sure to address the data and other policy-related requests noted at the end of this email.

IMPORTANT:

a) You will see that reviewer #1 raises two concerns. While we understand the reviewer's first point, and the additional cell culture experiment would further strengthen the paper, we and the Academic Editor will not insist on it; you may address this point as you feel fit.

b) Regarding the second point raised by rev #1, the Academic Editor says "Roscovitine and VX680 have different targets (CDK1 and Aurora kinase, respectively), so I would not expect the same phenotype from both treatments. I feel that the rationale of the authors is valid: if the phenotype was arising from loss of the positive feedback between CDK1 and Aurora kinase, then one would expect the same phenotype from inhibition of either kinase." As the other reviewers, who requested the additional inhibitor, are satisfied, we do not require additional data here.

c) Please attend to the requests from reviewer #3.

d) Please attend to my Data Policy requests further down.

e) Please could you change the title to something that clarifies which kinases and what type of receptors you study concerns. Maybe "Mitotic kinases CDK1 and Aurora choreograph storage and redistribution of membrane-bound receptors."

We expect to receive your revised manuscript within two weeks. Your revisions should address the specific points made by each reviewer. In addition to the remaining revisions and before we will be able to formally accept your manuscript and consider it "in press", we also need to ensure that your article conforms to our guidelines. A member of our team will be in touch shortly with a set of requests. As we can't proceed until these requirements are met, your swift response will help prevent delays to publication.

- a cover letter that should detail your responses to any editorial requests, if applicable

*Copyediting*

*Published Peer Review History*

*Early Version*

Sincerely,

Roli Roberts

Senior Editor,

rroberts@plos.org,

PLOS Biology

DATA POLICY:

Regardless of the method selected, please ensure that you provide the individual numerical values that underlie the summary data displayed in the following figure panels as they are essential for readers to assess your analysis and to reproduce it: Figs 2A’B’C’DEE’E’’E’’’FF’F’’F’’’ 3EHK, 4CHK, 5A’B’CFF’II’J, S1CDGH, S2AA’A’’A’’’, S3CDEFG, S4CDGH, S5A’B’C, S6ABCDGJKNO. NOTE: the numerical data provided should include all replicates AND the way in which the plotted mean and errors were derived (it should not present only the mean/average values).

REVIEWERS' COMMENTS:

Reviewer #1:

The authors did not satisfactorily address my point 1, beyond adding a few lines of text:

1. To make their findings more understandable to the non-specialist, the authors could consider converting part of the current Figure S1 into a revised Fig 1, to show where these founder cells are located within the embryo, and to show the cross-section of the wild type FGFR::Venus at interphase, metaphase and telophase, which is currently only schematized in Fig 1A. The authors should not assume that the reader has read and can recall their previous papers. Rather, they should feel comfortable in simply showing new versions of previously demonstrated data or

schematics again, at least of the wild-type cells and what happens to them during mitosis, so that the basic phenomenon is clearly established in Fig 1 without needing to read the text or previous papers. For example, why does the previous Dev Cell paper show a schematic of a the Ciona embryo founder cells and their location within the embryo in Fig 1A, while this new paper does not? It would also be worth showing new examples of the Fig 1B-D in the Dev Cell paper in this new paper. In particular, there is no interphase control cell shown for Fig 1B.

The authors responded that they do not need to address this point because "the current manuscript is focused on delineating the cell biology of mitotic trafficking, rather than heart progenitor induction". If this is the case, and their goal is general cell biology rather than the development of a specific cell type, then they need to show the same results in a cell culture system, such as human cells in culture, to confirm the generality of their findings to cell biology.

In their response to my point 4, the authors state:

"We are confident that feedback inhibition of CDK1 due to VX-680 treatment (or feedback inhibition of AurK due to roscovitine treatment) did not create artefactual results because treatment with these inhibitors (roscovitine vs. VX- 680) led to highly distinctive, non-overlapping effects on FGF receptor distribution and staining levels as detailed in Figures 3 and 5. For example, if VX-680 inhibited CDK1 then we would have expected to observe reduced FGFR:VENUS staining, instead VX-680 had no discernable impact on FGFR::VENUS staining levels (Figure 3)."

I am glad they are confident of their own interpretations, but peer review requires that an objective reviewer is also confident. The risk the authors face is that their results from either Roscovitine or VX680 are completely misleading and artefactual. And the fact that the two drugs have different effects only makes this concern greater, not less.

Reviewer #2:

The authors addressed all of my concerns and added new experiments that strengthened their conclusions and model.

I am fully satisfied and recommend publication. 

Reviewer #3:

In this revised version, Cota and colleagues have addressed nearly all my minor comments. In particular, the new AMG-900 data really does strengthen the manuscript, and I do agree with them that their orthogonal CyclinB(delta90) data is sufficient and that another CDK1 inhibitor is not required. I'd like to congratulate the authors on this very elegant piece of quantitative cell biology!

I have only two cosmetic comments:

-Regarding controls, I do agree that that the spatial relocalisation data is well controlled and convincing between Roscovitine and CyclinB(delta90) treatments (Fig.4A-C and S4A-C). My point was more for the specific experiment presented in Figure 3I. I personally would have put a picture of a live embryo expressing Mesp> LacZ; Mesp>FGFR::Venus (figure 3I) for the reader to directly see the point of the authors , even if it basically has noise in the Venus channel. But this is the author's choice, the added text and a "data not shown" convey the same idea.

-I thank the authors for clarifying their "a, b, ns" notation, it makes more sense now. To help the reader, the authors might want to add in the legend of Fig1c that the "a,b,ns" applies for changes *within the same region* between stages (i.e. not between two regions within a given stage, but within the same region between stage). Something like:

"Significant changes [within the same cell region] between stages (p<0.05) are indicated by a change in lettering (a, b), n.s.=not significant

---

## [Editor Report · Decision Letter 3]

18 Dec 2020

Dear Dr. Davidson,

I am writing concerning your manuscript submitted to PLOS Biology, entitled “Cyclin-dependent Kinase 1 and Aurora Kinase choreograph mitotic storage and redistribution of a growth factor receptor..”

We have now completed our final technical checks and have approved your submission for publication. You will shortly receive a letter of formal acceptance from the editor.

Kind regards,

PLOS Biology